https://doi.org/10.1038/s41467-020-19922-3　　OPEN

# Reproductive phasiRNAs regulate reprogramming of gene expression and meiotic progression in rice

Yu-Chan Zhang [1,3 ✉], Meng-Qi Lei[1,3], Yan-Fei Zhou[1], Yu-Wei Yang[1], Jian-Ping Lian[1], Yang Yu[1], Yan-Zhao Feng[1], Ke-Ren Zhou[1], Rui-Rui He[1], Huang He[1], Zhi Zhang[1], Jian-Hua Yang[1] & Yue-Qin Chen [1,2 ✉]

Plant spermatogenesis is a complex process that directly affects crop breeding. A rapid change in gene abundance occurs at early meiosis prophase, when gene regulation is selective. However, how these genes are regulated remains unknown. Here, we show that rice reproductive phasiRNAs are essential for the elimination of a specific set of RNAs during meiotic prophase I. These phasiRNAs cleave target mRNAs in a regulatory manner such that one phasiRNA can target more than one gene, and/or a single gene can be targeted by more than one phasiRNA to efficiently silence target genes. Our investigation of phasiRNA-knockdown and *PHAS*-edited transgenic plants demonstrates that phasiRNAs and their nucleotide variations are required for meiosis progression and fertility. This study highlights the importance of reproductive phasiRNAs for the reprogramming of gene expression during meiotic progression and establishes a basis for future studies on the roles of phasiRNAs with a goal of crop improvement.

[1] Guangdong Provincial Key Laboratory of Plant Resources, State Key Laboratory for Biocontrol, School of Life Science, Sun Yat-Sen University, Guangzhou 510275, P. R. China. [2] Guangdong Laboratory of Lingnan Modern Agriculture, Guangzhou 510642, China. [3] These authors contributed equally: Yu-Chan Zhang, Meng-Qi Lei. ✉email: zhyuchan@mail.sysu.edu.cn; lsscyq@mail.sysu.edu.cn

Meiotic entry is a complex process, and cell morphology and gene expression change drastically during pre-meiotic and early meiotic stages. In plants, meiotic progression directly affects crop breeding and agricultural yields; therefore, understanding the mechanism of meiotic progression is important. Recently, the timing and gene expression profiles of the events associated with these processes were reported in maize, and large programmed changes in gene expression at the start of meiotic prophase have been identified[1]. However, how these genes are regulated remains to be clarified. Notably, a group of 21-nucleotide (nt) phased secondary siRNAs (phasiRNAs) were reported to exhibit a burst of expression in grass inflorescences from the premeiosis stage and to decrease in expression at the end of meiosis, when extensive gene expression changes occur[2–8]. These phasiRNAs are processed from precursors that contain miR2118-binding sites and might be triggered by miR2118[2,9]. More importantly, a differential accumulation of phasiRNAs is associated with photoperiod-sensitive male sterility in rice[10]. Collectively, these lines of evidence suggest a link between 21-nt phasiRNAs and global gene regulation during early meiosis.

Small RNAs (sRNAs) usually play a role in RNA silencing or epigenetic regulation by binding and guiding AGO proteins to their specific targets[11]. Although some phasiRNAs operate in *cis* and target their own precursors via perfect matches[12,13], phasiRNAs have no obvious *trans* targets, which makes it difficult to predict their authentic target genes. In rice, 21-nt phasiRNAs were reported to bind an AGO protein, MEIOSIS ARRESTED AT LEPTOTENE1 (MEL1), and loss of MEL1 function induces the arrest of meiosis at prophase and leads to anomalous pollen mother cells (PMCs)[4,14,15]. We hypothesize that 21-nt phasiRNAs could silence a specific set of genes and be associated with global gene regulation during early meiosis. To test this hypothesis, it is necessary to identify the targets of these reproductive phasiRNAs. Furthermore, experiments with loss- or gain-of-function mutants are needed to characterize the target preference and cleavage patterns of the phasiRNAs.

In this study, we generate *MEL1* loss-of-function mutants (*mel1*) and collect young spikelets of wild-type (WT) and *mel1* plants at different developmental stages for multi-omics sequencing of sRNAs, the degradome and transcriptome. Our results show that MEL1–phasiRNAs are essential for *trans* RNA cleavage and mRNA elimination during meiotic prophase I. Moreover, analysis of a transgenic plant series reveals that the phasiRNAs and their nucleotide sequences are important for their function and for the progression of meiosis. Our results thus highlight the essential functions of reproductive phasiRNAs in rice during meiosis progression and fertility.

## Results

**Reproductive phasiRNAs are involved in RNA cleavage during the early stages of PMC meiosis.** To identify phasiRNAs expressed in the inflorescence that might form the siRNA–AGO complex during meiotic prophase I and that have the potential to participate in mRNA cleavage, we first sequenced sRNAs from spikelets of ZH11 (Zhonghua 11, *Oryza sativa japonica*) in three biological replicates at four different sporogenesis stages: the PMC Formation Stage (PFS), PMC Prophase Stage (PPS), PMC Meiotic Divisions Stage (PMDS), and Early Microspore Stage (EMS) (Supplementary Data 1). We also created *MEL1* loss-of-function mutants (*mel1*) by CRISPR-Cas9 (Supplementary Fig. 1) for validation and comparison. Because the *mel1* PMCs arrested in prophase[14], young spikelets of *mel1* plants at the PPS were collected (pipeline shown in Supplementary Fig. 2). Reproductive phasiRNAs that were highly expressed at the PFS and significantly ($p < 0.05$) decreased in expression at the EMS were

extracted for subsequent study (Supplementary Fig. 3a; Supplementary Data 2). About one third of reproductive phasiRNAs were MEL1-bound (DDBJ: DRP000161)[4]. Approximately 64.9% of reproductive phasiRNAs originated from intergenic regions (Supplementary Fig. 3b).

siRNAs might be involved in translational regulation or RNA cleavage in both plants and animals[16,17]. To understand in which process the MEL1–phasiRNAs function, we firstly separated polyribosomes (polysomes) and detected the subcellular localization of MEL1. The MEL1 protein is present in the cytosol, but not in total cellular polysomes isolated from the WT PFS spikelet (Fig. 1a), indicating that it is unlikely to function in translational regulation. We also analyzed the subcellular distribution of four phasiRNAs with varying abundances. Cytoplasmic extracts were prepared from WT PFS spikelets and fractionated on sucrose gradients (Fig. 1b). As positive controls, we used *GAPDH* and miR171[18,19], which have been reported to be present in supernatant and polysomal fractions in *Arabidopsis* (Fig. 1c). qRT-PCR revealed that all phasiRNAs were present in the supernatant fraction but not in the polysomal fractions, whereas *GAPDH* existed in polysomal fractions and miR171 was more enriched than phasiRNAs in the polysomal fractions, further confirming that they might not be involved in translational regulation (Fig. 1c). Together with the general cleavage capacity of the AGO–siRNA complex in plants, these results suggest that MEL1–phasiRNAs preferentially function by post-transcriptionally silencing RNAs but not by inhibiting their translation.

We next screened the cleavage targets of phasiRNAs using target prediction and PARE sequencing target validation (Supplementary Data 1). It has been speculated that the degree of complementarity of 21-nt reproductive phasiRNAs to their targets is less than that for miRNAs[12]. Thus, target genes were retained for further analysis that: (1) paired with phasiRNAs with mis-pairing scores ≤ 5; (2) had a threshold corrected $p$-value $< 0.05$; (3) had a normalized PARE read abundance ≥ 4; and (4) had a PARE read abundance above the median for the gene. In total, 5040 candidate protein-encoding target genes were identified. The phasiRNAs also cleaved 3103 transposable elements (TEs) (Supplementary Fig. 3c). To reduce the false-positive targets, we further applied transcriptome sequencing for each sample with three biological replicates to establish the expression patterns of the potential target genes (Supplementary Data 1). Reproductive phasiRNAs have been reported to be loaded into MEL1 (at least partially) to perform their function, and the mRNA abundance in samples in which mRNA cleavage occurred should therefore not be higher than that in samples with no cleavage. We then filtered out the targets that were cleaved in WT samples but were not cleaved in *mel1*, and whose expression was downregulated in *mel1*. The remaining cleaved target genes were considered to be MEL1-dependent target genes and were used for the subsequent analysis (Supplementary Data 3). The cleaved targets that were filtered out might be MEL1-independent target genes (Supplementary Data 4). In total, 2281 targeted protein-coding genes were identified. In addition, 2288 phasiRNAs (20.5% of the total identified reproductive phasiRNAs) were predicted to have targets, and these phasiRNAs derived from 1349 PHAS loci. The expression of most of these cleaved targets was downregulated upon entry into meiosis (Fig. 1d). Most of the phasiRNAs functioned in *trans* and did not preferentially target neighboring genes (Supplementary Fig. 3d). The phasiRNAs preferentially cleaved the coding sequence region near the stop codon and the 3′-untranslated region (UTR) of target genes (Fig. 1e).

**Reproductive phasiRNAs cleave targets to eliminate specific RNAs that potentially negatively regulate sporogenesis.** We next analyzed the composition of the 2841 cleavage targets of

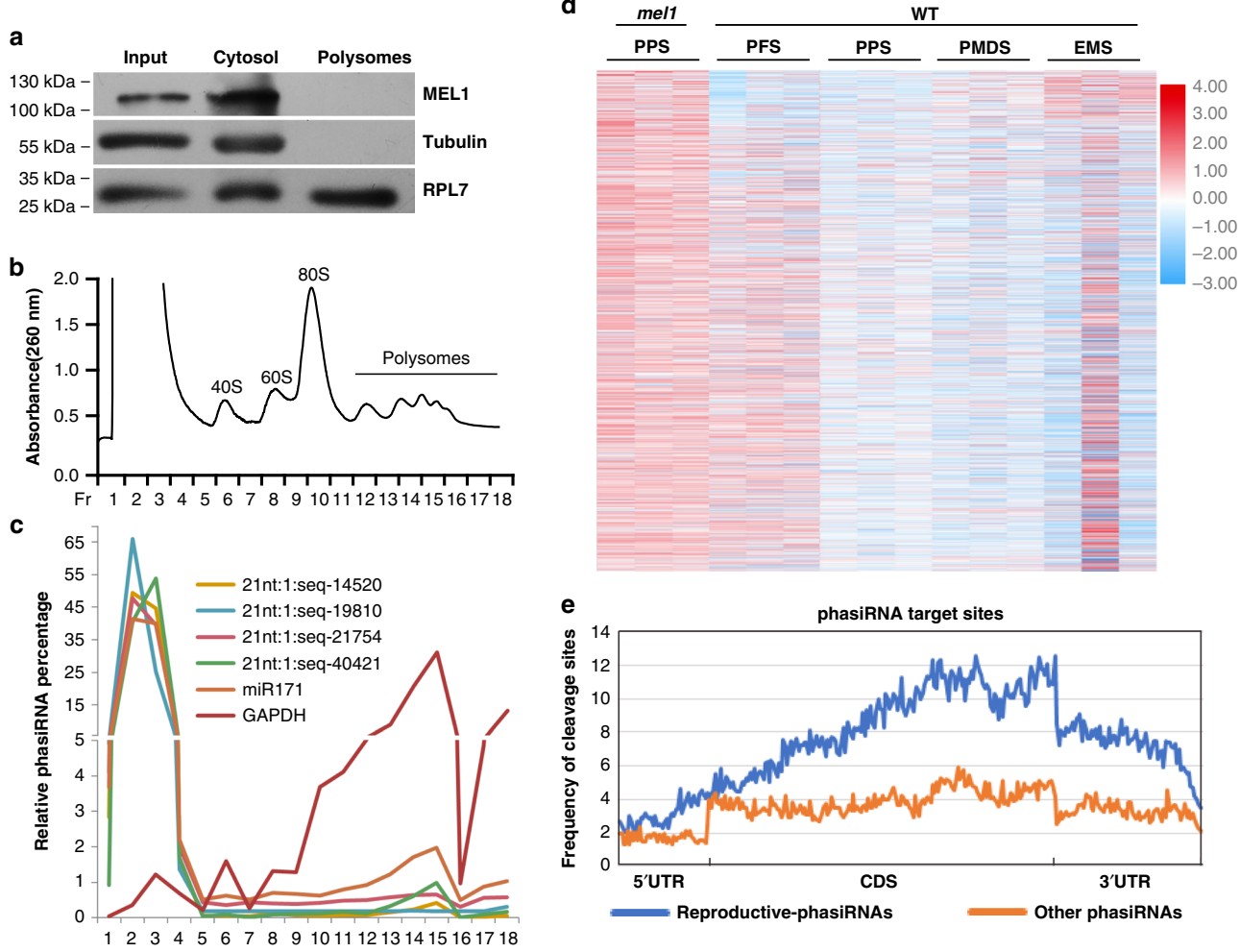

**Fig. 1 Analysis of MEL1-dependent phasiRNAs and their target preference and rules. a** Western blot analyses the distribution of MEL1 protein. 60s ribosomal protein L7 (RPL7) and tubulin were used as controls. Uncropped blots in Source Data. $n = 3$ independent replicates. **b** Sucrose-gradient analysis of extracts from wild-type (WT) PMC premeiosis-stage spikelets. Fractions 1–5 are the supernatant; fractions 6–11 are the 40s–80s; fractions 12–18 are polysomes. **c** The distribution of phasiRNAs and two controls (miR171 and *GAPDH*) in each sucrose-gradient fraction were quantified by qRT-PCR. Relative distribution of the RNA in each fraction was normalized by the total abundance of RNA in all fractions marked as 100%. $n = 3$ independent replicates. **d** Heatmap of all target coding genes of reproductive phasiRNAs in WT and *mel1* during anther development. Gene expression values used to create the heatmap are the log_2FPKM (Fragments Per Kilobase Million) of the target genes of reproductive phasiRNAs, and the values were scaled by row. Abbreviations: PFS PMC formation stage, PPS PMC prophase stage, PMDS PMC meiotic divisions stage, EMS early microspore stage. **e** The distribution of phasiRNA-targeting sites on the target transcripts is divided into 5′-UTR, coding sequence, and 3′-UTR.

reproductive phasiRNAs. In contrast to the targets of piwi-interacting RNAs (piRNAs), those of phasiRNAs are apparently biased. We compared the significantly enriched GO terms of phasiRNA targets with those previously reported to be enriched among differentially expressed genes identified by single-cell sequencing during maize meiosis[1], to investigate the association of phasiRNAs and the timing and gene expression profiles of premeiosis and meiosis events. The enriched GO terms for mRNAs targeted by phasiRNAs had a 55.6% overlap with those of the reported cluster 1 genes in maize, which represent rapidly downregulated genes[1], but hardly overlapped (3.7%) with those (clusters 2–6) that consisted of more slowly up- or downregulated genes (Supplementary Data 5). This pattern indicates that the rapid downregulation of genes in specific categories during meiosis is probably initiated by phasiRNAs.

The phasiRNA target mRNAs were significantly enriched for the GO terms adenyl ribonucleotide binding (19.8%), kinase activity (12.1%), and hydrolase activity (5.2%) (Supplementary Fig. 4; Supplementary Data 5). Among the targeted kinase genes,

71.9% encode dual-specificity protein kinases (DPKs), and 72.2% of the targeted DPKs are receptor-like kinases (RLKs) (Fig. 2a). It has been suggested that plants use hundreds of receptors as entry points into signaling pathways for both developmental and environmental responses[20]. The phasiRNA-targeted RLKs mainly consist of leucine-rich-repeat (LRR) RLKs and S-domain family RLKs (Fig. 2a). Some LRR-RLKs, such as MULTIPLE SPOR-OCYTE (MSP1)[21] in rice, regulate early floral organ development and are greatly downregulated in microsporocytes during entry into meiosis[22–26]. The S-domain family RLKs determine self-incompatibility responses, and rejection of pollen occurs when S-RLKs are present in both pollen and female receptive tissues[27,28]. In addition to the kinase genes cleaved by phasiRNAs, several genes related to transcription and RNA metabolism were also enriched among the GO terms (Supplementary Fig. 4; Supplementary Data 5). A significant global upregulation of genes was revealed in *mel1* samples compared with WT samples at the PPS (Fig. 2b), indicating that MEL1 is indispensable for the reprogramming of mRNA expression during early meiosis.

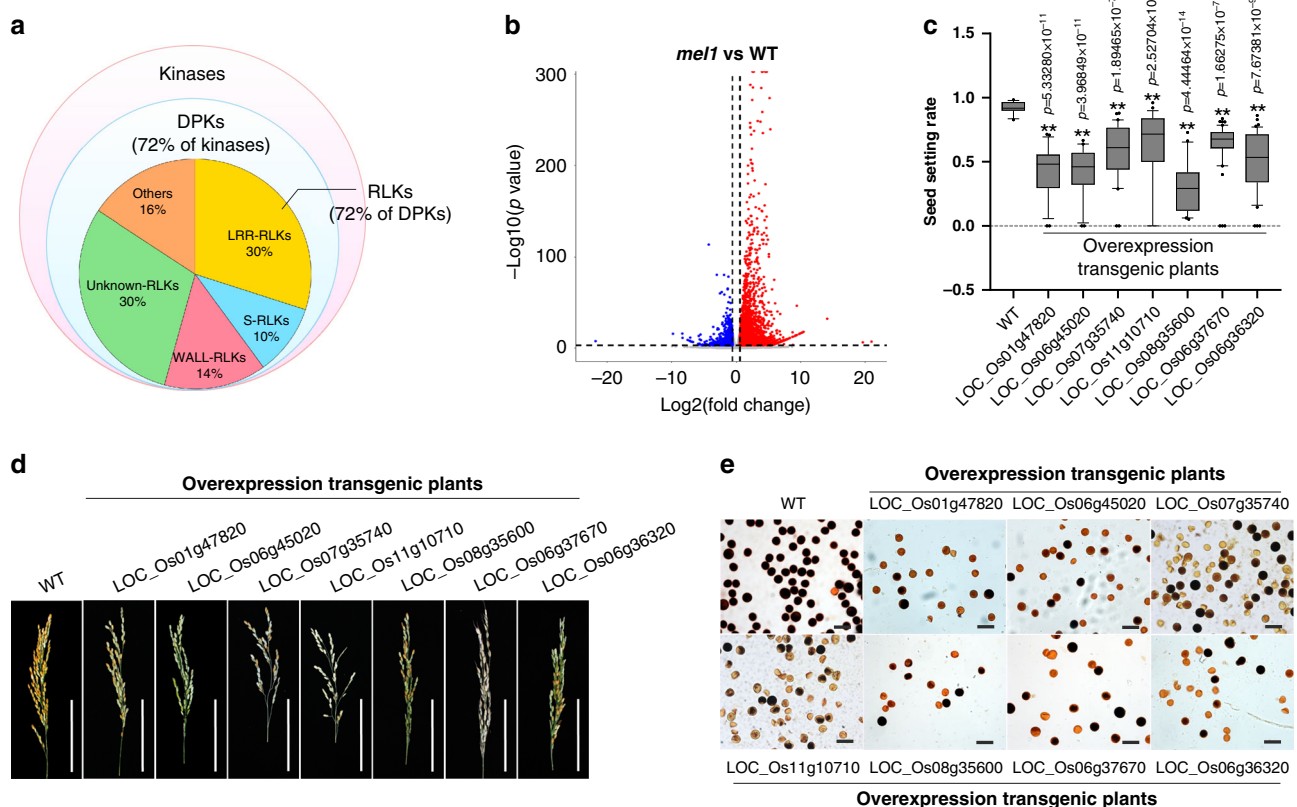

**Fig. 2 Reproductive phasiRNA target genes that negatively regulate sporogenesis. a** Pie charts showing the categories of the kinases targeted by reproductive phasiRNAs. **b** Volcano plots of the global gene expression pattern in *mel1* and wild-type (WT) spikelets at the PPS. Red spots represent genes that upregulated in *mel1*, and blue spots represent genes that downregulated in *mel1*. **c** Seed-setting rate of WT and different transgenic plants. Each target gene was overexpressed independently. Box plots indicate median (middle line), 25th, 75th percentile (box) and 10th and 90th percentile (whiskers) as well as outliers (single points). ($n = 15, 23, 20, 38, 26, 23, 50$, and 42 plants from left to right, respectively). Significant differences were identified at the 5% (*) and 1% (**) probability levels using two-tailed paired *t*-test. **d** Panicles of WT and different transgenic plants. Scale bars = 10 cm. **e** Pollen grains stained with iodine-potassium iodide of the WT and different transgenic plants. Scale bars = 100 µm. $n = 3$ independent replicates.

Because the target gene functions of most reproductive phasiRNAs are unknown, we then randomly chose seven of the target genes and created transgenic plants that overexpressed these genes using ubiquitin promoter, to evaluate the effect of high expression levels of these target genes on sporogenesis and fertility (Fig. 2c–e and Supplementary Fig. 5a). The requirement of phasiRNAs, the phasiRNA-binding sequence, and MEL1 for the downregulation of two of the target genes was further validated using a rice protoplast transient transformation system (Fig. 3a). We also analyzed transgenic plants that overexpressed non-phasiRNA-targeted genes, including genes that are either expressed or not expressed in spikelets as negative controls (Supplementary Fig. 5b, c). Almost all of the T₂ transgenic plants displayed sterility or semi-sterility phenotypes (Fig. 2c, d), and most of the pollen grains were aborted (Fig. 2e). None of these phenotypes was observed in negative control plants (Supplementary Fig. 5d, e), suggesting that these target genes negatively affect sporogenesis and fertility. To demonstrate further that the semi-sterile phenotype of plants overexpressing the target genes was caused by pollen defects, we crossed these plants with WT plants. When the mutants were used as a male parent, the rate of seed set decreased compared with that using WT as a male parent and the mutants as female parents (Supplementary Fig. 6a, b), indicating that the pollen-grain defects in these mutants impaired pollination and seed set. Thus, phasiRNAs eliminate specific RNAs that potentially negatively regulate sporogenesis.

We next analyzed the cleavage principle of phasiRNAs and observed that 18.6% of phasiRNAs cleaved more than one target gene belonging to different gene families, and 15.7% of target genes were cleaved by more than one phasiRNAs from different *PHAS* loci (Supplementary Data 3). This targeting approach might improve the silencing efficiency of target genes and increase error tolerance. No apparent chromosomal preference of a phasiRNA for its target was observed (Supplementary Fig. 6c). We further used 5′ rapid amplification of cDNA ends (5′-RACE) to validate cleavage by phasiRNAs. Cleavage sites that either had a high number of PARE reads (15–180) or a low number of PARE reads (<10) were chosen randomly for RACE validation, and 10 out of 15 chosen cleavage sites were successfully validated (Fig. 3b).

**Reproductive phasiRNAs are required for meiotic prophase progression and male fertility in rice.** We then asked whether the loss of phasiRNAs also affects meiotic progression. Because reproductive phasiRNAs are transcribed from *PHAS* loci and processed by DCL4 and miR2118[4,29], we therefore obtained DCL4- and miR2118-knockdown plants (*dcl4* and *mir2118*) and constructed a series of mutants of two *PHAS* loci for analysis. Similar to *mel1* plants, *dcl4* plants were sterile and produced no mature pollen grains; however, the *mir2118* plants were semi-sterile, and approximately half of the pollen grains were aborted

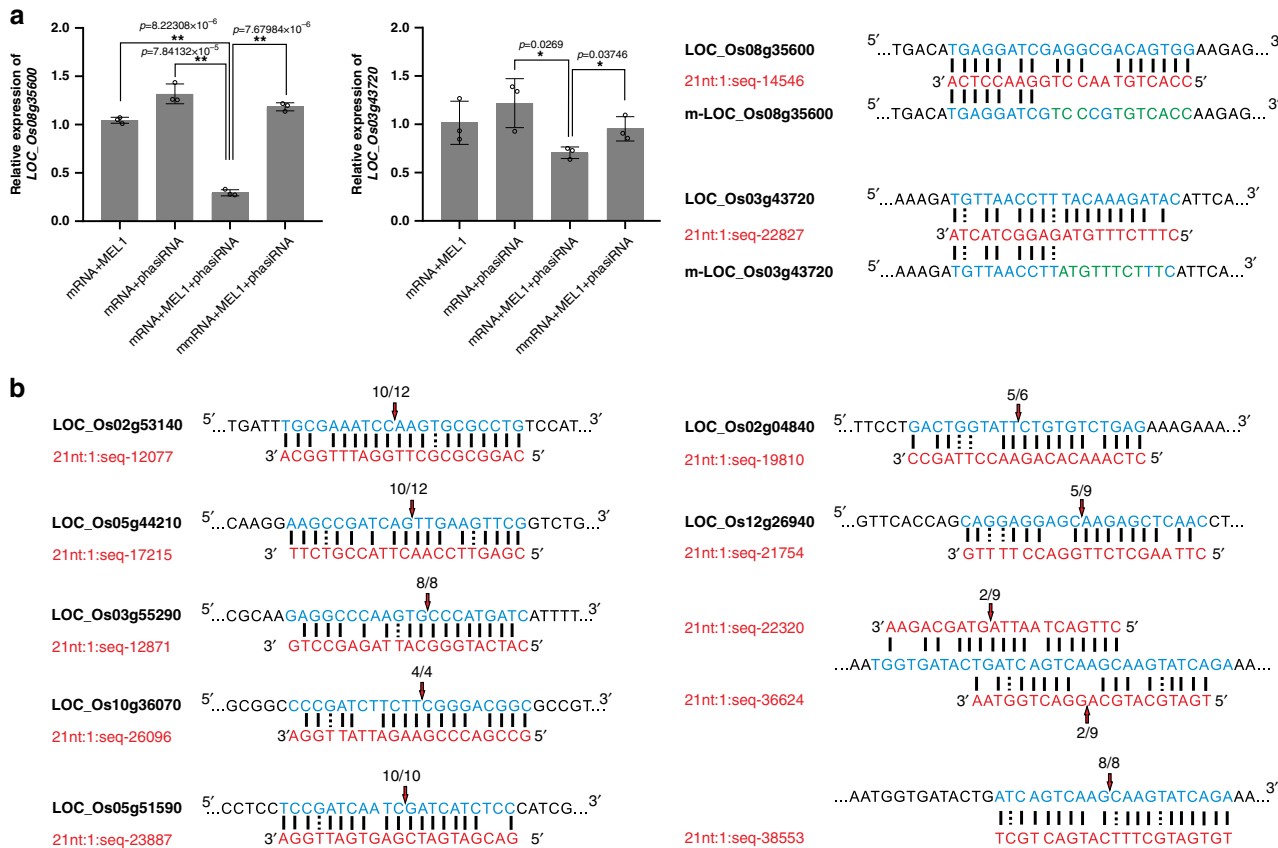

**Fig. 3 Validation of phasiRNA target genes. a** Validation of the requirement for phasiRNAs, phasiRNA-binding sequence and MEL1 for the downregulation of target genes in a rice protoplast transient transformation system and analysis by qRT-PCR. Values are means ± SD ($n = 3$ replicates). The phasiRNAs are marked in red, binding sites are marked in blue, and the mutated sequences of the binding sites are marked in green. Significant differences were identified at the 5% (*) and 1% (**) probability levels using the two-tailed paired $t$-test. **b** 5′- RACE verification of the cleavage of seven target genes by different phasiRNAs. PhasiRNAs are marked in red, binding sites are marked in blue, the arrows represent the cleavage site, and the numbers above the binding sites indicate the number of the cleavage products matched at the site to that of the total cleavage products.

(Fig. 4a). The difference between the phenotypes of *mir2118* plants and those of *mel1* and *dcl4* plants might result from the incomplete knockdown of miR2118 (Supplementary Fig. 7a), because the miR2118 cluster contains 18 members with different sequences in the rice genome (annotated by miRBase).

We next investigated the process of meiosis in *dcl4* and *mir2118* plants and observed that meiosis was completely disordered in *dcl4* plants. We observed an apparent arrest of *dcl4* PMCs at prophase I compared with the PMCs of WT plants, and at the microspore stage, 62.7% of *dcl4* PMCs were arrested at prophase I (Fig. 4b). Moreover, we observed abnormal *dcl4* PMCs that possessed chromosome bridges and conglutination, and abnormal cell division (Fig. 4c). The meiosis phenotype was moderate in *mir2118* plants, consistent with abortion of about half of the pollen grains (Fig. 4c). These results indicated that phasiRNAs are required for meiosis progression, especially that of meiosis prophase I.

A single-nucleotide polymorphism near the miR2118 recognition site in a potential phasiRNA precursor, *PMS1T*, might affect phasiRNA accumulation and sporogenesis[10]. Sequence variation is important for crop domestication[30,31]; therefore, we analyzed the effects of *PHAS* locus mutations on rice male fertility by mimicking natural variation via mutation of one or more nucleotides near the potential miR2118 binding sites in two independent reproductive phasiRNA precursor lncRNAs, *MSPPL1* and *MSPPL2* (*Male Sterility-related PhasiRNA Precursor LincRNAs*). The two lncRNAs were selected on the basis of the

anther-specifically expressed lncRNAs that we identified previously[32] and the *PHAS* loci with potential miR2118 binding sites that we identified in this study. The two lncRNAs with the highest expression level were then chosen and their gene structure and information are shown in Supplementary Fig. 7b. To monitor the importance of the sequence up- or downstream from the trigger sites on phaiRNA function, we mutated two sites within the *PHAS* loci either up- or downstream from the potential miR2118 binding sites by CRISPR/Cas9 (Fig. 5a and Supplementary Fig. 7b). Two phasiRNAs near the *MSPPL1* edited sites and four phasiRNAs near the *MSPPL2* edited sites were detected by sRNA-Seq.

We generated transgenic lines harboring different mutations at the target site (*msppl1* and *msppl2*). Six *msppl1* mutants were used in this study, including: 1) Deletion of the miR2118 binding site (*miR2118 del*); 2) edited sequences upstream from the potential miR2118 binding site (but that did not affect the phasiRNA sequence or miR2118-binding site) (*PAM1 edit -1 and -2*); 3) edited sequences both up- and downstream from the potential miR2118-binding site (that affected phasiRNA sequence but did not affect the miR2118-binding site) (*PAM1&2 edit -1, -2 and -3*) (Supplementary Fig. 7b). Seven *msppl2* mutants were used in this study and all were edited both up- and downstream from the potential miR2118-binding sites, but contained different edited sequences (that affected the sequences of the phasiRNAs 21nt:1: seq-22827 and 21nt:1:seq-33993 but did not affect miR2118-binding site) (*PAM2 ≥ 5 bp del -1, -2 and -3; PAM2 4 bp del -1 and*

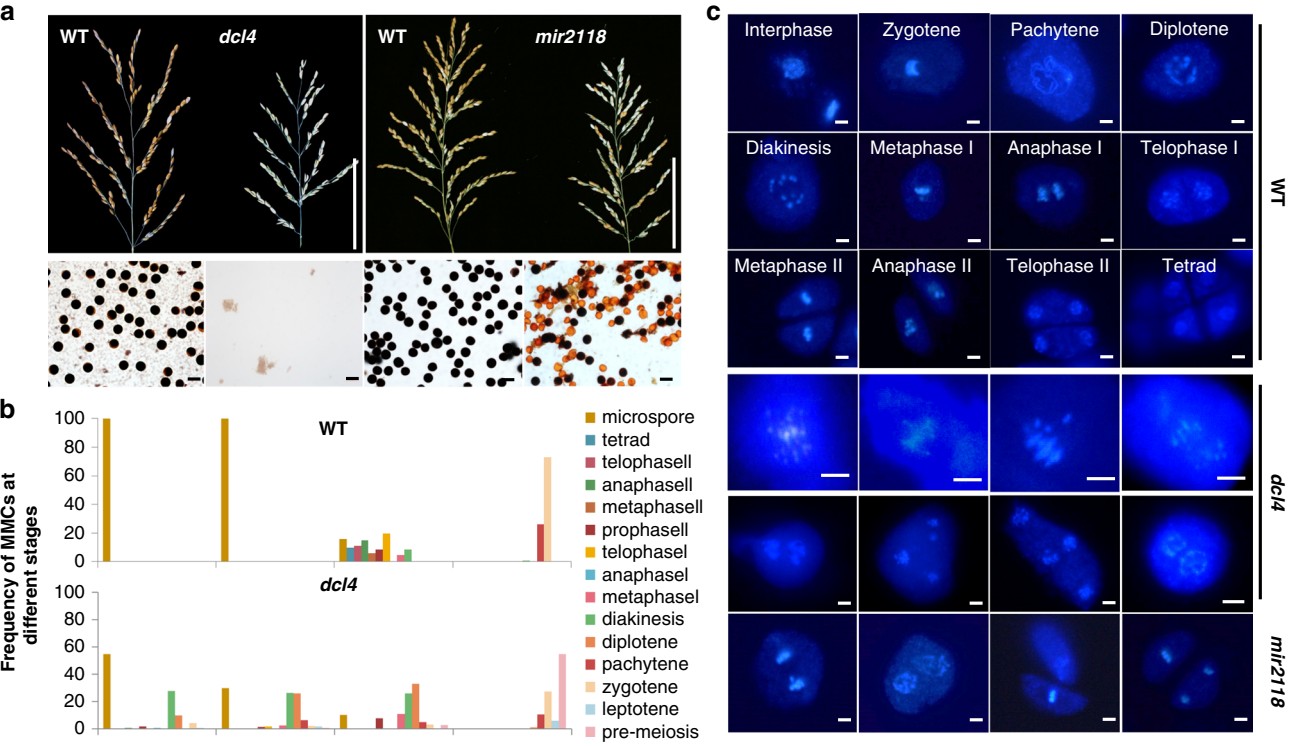

**Fig. 4 PhasiRNAs are required for meiosis progression. a** Panicles and pollen grains (stained with iodine-potassium iodide) of *dcl4*, *mir2118*, and wild-type (WT) plants. Scale bars = 10 cm for panicles and 100 μm for pollen grains. *n* = 4 independent replicates. **b** Frequency of PMCs at various meiotic stages in anthers ranging from 0.7 to 1.1 mm in length. 84–373 pollen mother cells were accessed from approximately 30–40 anthers for each anther size class, and these were collected from five *dcl4* transgenic plants and ten WT plants. **c** The meiosis processes by staining with 4′,6-diamidino-2-phenylindole (DAPI) of WT, *dcl4*, and *mir2118* PMCs. Scale bars = 4 μm. *n* = 5 independent replicates.

-2; *PAM2 3 bp del -1* and *-2*) (Supplementary Fig. 7b). Most of the mutations caused semi-sterile phenotypes to varying degrees but did not affect vegetative growth (Fig. 5b, c). The decreased fertility was evident by the presence of aborted pollen grains, which was further supported by hybridization experiments (Fig. 5d and Supplementary Fig. 6a). The loss of function of one *PHAS* locus did not cause serious defects in seed setting, probably because either two phasiRNAs function in concert, the targeted genes are not relevant, or no change in targeting occurs. We analyzed the effects of the mutation site and the degree of sequence variation on the rate of seed set. When the two sites upstream and downstream from the miR2118-binding site were both mutated, the rate of seed set decreased the greatest and the deletion of more nucleotides resulted in an even greater decrease in seed set (Fig. 5c). These results showed that the degree of sequence integrity at these two *PHAS* loci is essential for rice male fertility.

**Nucleotide variation within phasiRNA precursors affects phasiRNA sequences and cleavage of their targets.** Finally, we investigated how sequence variation affects the post-transcriptional regulatory role of phasiRNAs on gene expression. Gene editing up- and downstream of the miR2118-binding site in the *msppl1* mutants downregulated phasiRNA expression and slightly upregulated that of precursor lncRNAs (Supplementary Figs. 7c and 8a). Deletion of the miR2118-binding site upregulated *MSPPL1* expression and led to the absence of phasiRNAs (Supplementary Figs. 7c and 8a). The detected phasiRNA targets were expressed in an almost complementary manner to their targeting phasiRNAs (Supplementary Fig. 8b). Moreover, the mutation of nucleotides downstream from the miRNA-

binding site generated phasiRNAs that harbored the edited sequences (21nt:1:Seq-23167[ed]. and 21nt:1:Seq-41154[ed].), which could introduce new regulatory target pairs (Supplementary Fig. 8a). We chose four predicted targets for each phasiRNA[ed]. that had the highest expected values, and detected apparent downregulation of seven of them in two independent lines of *msppl1* (PAM2 edited) (Supplementary Fig. 8c).

For *MSPPL2*, four phasiRNAs near the editing sites were predicted to target four protein-coding genes (Fig. 5a) according to the degradome and transcriptome data (Supplementary Data 3). Similar to the *MSPPL1* mutants, mutations led to a slight upregulation in *MSPPL2* expression (Supplementary Fig. 7c), but the expression of three out of the four detected phasiRNAs was significantly downregulated (Fig. 5e), and the detected phasiRNA targets were expressed in an almost complementary manner to their targeting phasiRNAs (Fig. 5f). To analyze the function of its target genes on pollen development and seed setting, we then constructed transgenic plants that overexpressed these target genes. An increase in the abundance of these target genes reduced pollen-grain fertility and seed setting rate (Fig. 5g–i). Collectively, the results showed that sequences at the two chosen *PHAS* loci are essential for the post-transcriptional regulatory roles of phasiRNAs and for male fertility, which might be important for crop domestication.

In conclusion, we showed that at the start of meiosis, a specific group of genes is post-transcriptionally regulated by a set of reproductive phase-specific phasiRNAs. The target preference of these phasiRNAs differ from that of most plant sRNAs, which is important for meiosis progression. A working model for the role of phasiRNAs in meiosis regulation is shown in Fig. 6. Using transgenic plants, we also experimentally tested how sequence

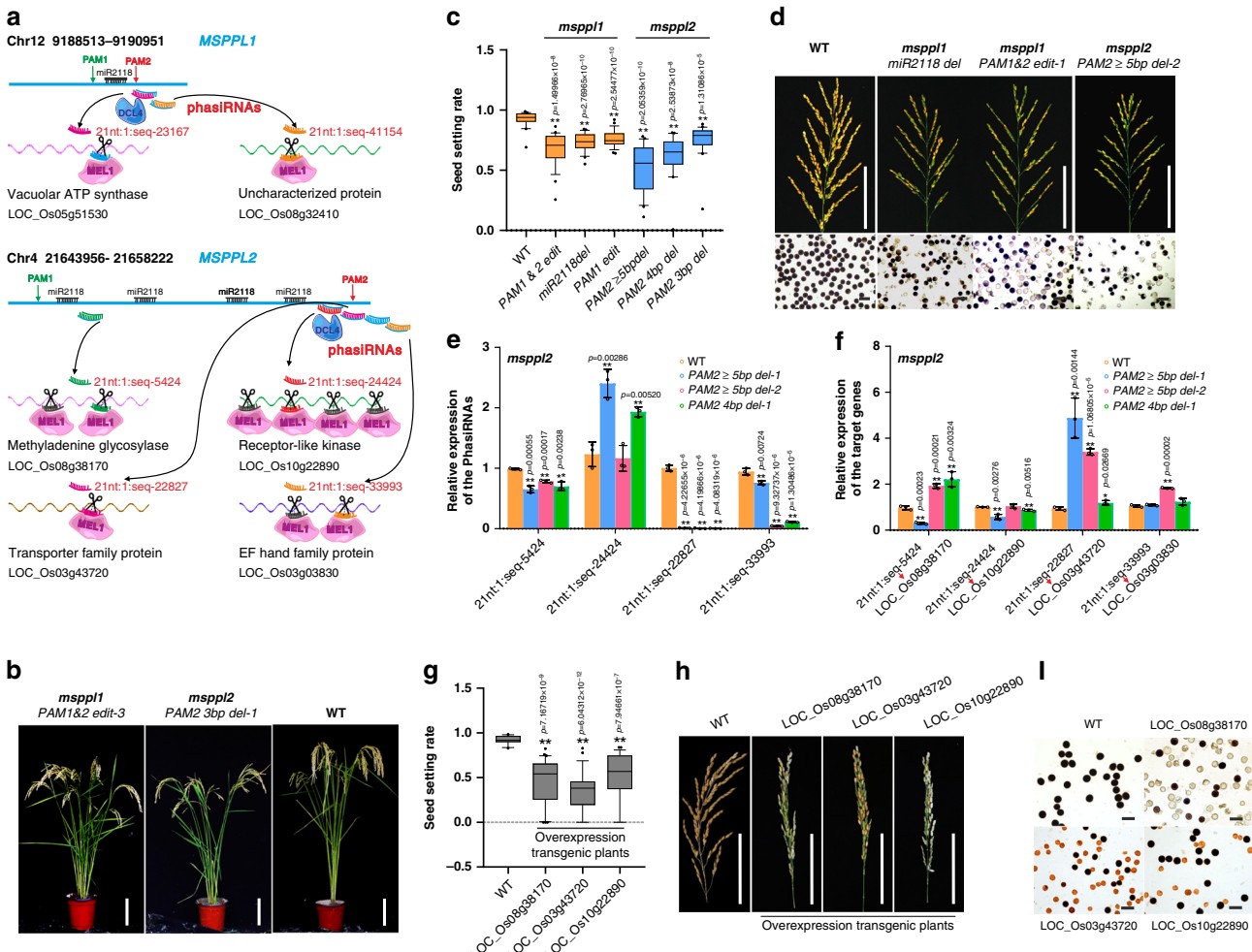

**Fig. 5 Functional analysis of phasiRNAs generated from *MSPPL1* and *MSPPL2*. a** Diagram of the two phasiRNA precursor lncRNAs, phasiRNAs and their targets, as well as the knockout mutants generated by CRISPR-Cas9 with PAM sites for Cas9-binding. The red and green arrows indicate the two gRNA target sites. **b** Gross morphology of wild-type (WT) and *msppl1* and *msppl2* mutant plants. Scale bars = 15 cm. **c** Seed-setting rate in different mutants of *MSPPL1* and *MSPPL2* (the data were generated from *PAM1&2 edit -1, -2, -3; PAM1 edit -1, -2, PAM2 ≥ 5 bp del -1, -2, -3; PAM2 4 bp del -1, -2* and *PAM2 3 bp del -1, -2* lines, respectively). Box plots indicate median (middle line), 25th, 75th percentile (box) and 10th and 90th percentile (whiskers) as well as outliers (single points). (n = 20, 23, 22, 26, 20, 20, and 21 plants from left to right, respectively). **d** Pollen grains and panicles of different mutants of *MSPPL1* and *MSPPL2*. Scale bars = 10 cm for panicles and 100 μm for pollen grains. n = 3 independent replicates. **e** Relative expression of phasiRNAs generated from *MSPPL2*. Values are the means ± SD (n = 3 replicates, normalized against *u6*). **f** Relative expression of target genes of phasiRNAs generated from *MSPPL2*. Values are the means ± SD (n = 3 replicates, normalized against *Actin2*). **g** Seed-setting rate of WT and different overexpression transgenic plants. Each target gene was overexpressed independently. Box plots indicate median (middle line), 25th, 75th percentile (box) and 10th and 90th percentile (whiskers) as well as outliers (single points). (n = 15, 30, 28, and 29 plants from left to right, respectively). **h** Panicles of WT and different overexpression transgenic plants. Scale bars = 10 cm. **i** Pollen grains of WT and different overexpression transgenic plants stained with iodine-potassium iodide. Scale bars = 100 μm. n = 3 independent replicates. Significant differences were identified at the 5% (*) and 1% (**) probability levels using two-tailed paired *t*-test.

variation within reproductive phasiRNAs affects their post-transcriptional regulatory roles and demonstrated the importance of phasiRNAs and the sequences at *PHAS* loci with respect to phasiRNA function, meiosis, and fertility in rice plants. This study reveals a regulatory pathway during rice sporogenesis that might be conserved in other important crop species.

## Discussion

Gene expression is precisely regulated during spermatogenesis, and the reprogramming of gene transcription during spermatogenesis necessitates the post-transcriptional regulation of the spermatogenic transcriptome. Several RNA degradation pathways are acutely important for male germ cell development, such as those piRNAs involved in post-transcriptionally silencing LINE1

elements[33] and transcriptome clearance during spermiogenesis in animals[34]. At meiotic prophase I, two steps of gene expression transition occur: one within the leptotene stage and the other at the entry into the prezygotene stage, and during this transition, a large number of genes are downregulated[1]. However, the responsible underlying mechanism for this is poorly understood in plants. During the transition to the rapid expression of mRNAs at prophase I, the expression in the inflorescence of a group of 21-nt phasiRNAs within intergenic regions is activated, and the loss of function caused by their binding to the AGO protein MEL1 arrests meiosis at the leptotene stage, which is the first step in the gene expression transition. However, the physiological function of these reproductive phasiRNAs is unclear. Here, we show that MEL1–phasiRNAs are essential for the reprogramming of gene expression during meiotic prophase I in rice.

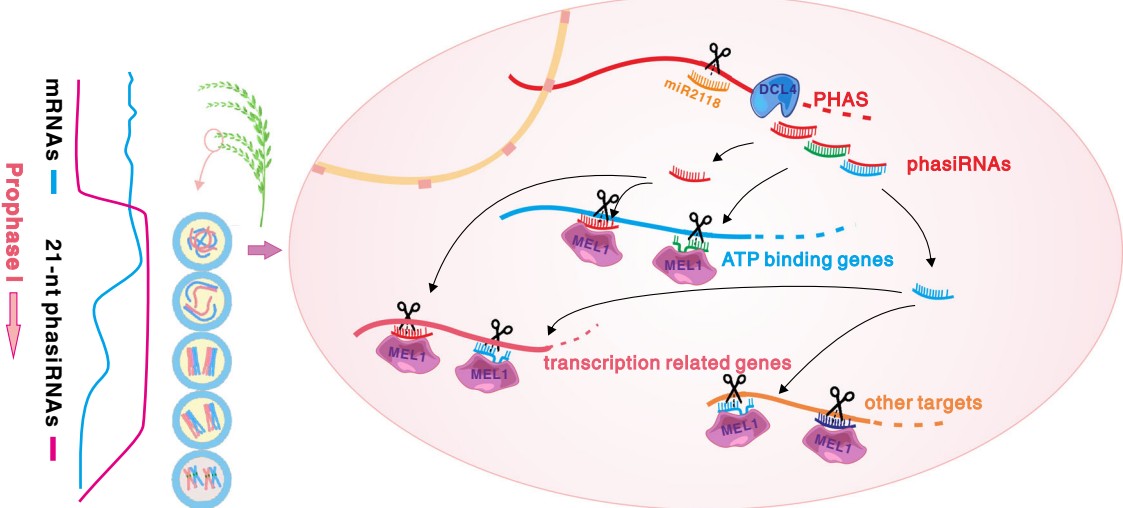

**Fig. 6 A working model for reproductive phasiRNAs which cleave and regulate target genes during meiotic prophase I in rice.** 21-nt phasiRNAs exhibit a burst of expression in inflorescences from the premeiosis stage and to decrease in expression at the end of meiosis, when extensive gene expression changes occur. They are essential for trans RNA cleavage and mRNA elimination during meiotic prophase I. The phasiRNAs prefered to target to ATP binding genes and transcription related genes.

First, we characterized the cleavage principle of reproductive phasiRNAs and showed that the target preference of the reproductive phasiRNAs differs from that of most plant sRNAs. Most of the reported non-reproductive phasiRNAs in dicotyledons preferentially target genes in a limited number of families, including those that encode MITE transposons and genes of the *PPR*, *NB-LRR*, *ARF*, and *MYB* families[35–37]. However, we show here that rice reproductive phasiRNAs have a much broader range of targets and preferentially cleave transcripts that belong to the rapidly downregulated genes during both steps of the expression transition at prophase I[1]. In addition, almost all reported plant sRNAs cleave their targets on the basis of near-perfect sequence complementarity, but reproductive phasiRNAs can tolerate several sequence mismatches when recognizing and cleaving their target genes, which supports the hypothesis that 21-nt reproductive phasiRNAs are less complementary to their targets than miRNAs[12]. This effect might be due to different evolutionary pressures on reproductive phasiRNAs and miRNAs during development. Notably, although phasiRNAs in rice and piRNAs in mice can eliminate mRNAs during sporogenesis, they function according to different working models: piRNAs in animals are mainly involved in the non-selective elimination of mRNAs at the late spermatogenesis stage at which meiosis has already finished[34], whereas rice phasiRNAs eliminate specific categories of genes during meiotic prophase I. In mice, mRNA elimination during meiotic prophase I might be regulated by other mechanisms, such as mRNA uridylation[38]. It is intriguing that plants and animals might have evolved different mechanisms to regulate genes during meiotic prophase.

Second, in addition to the cleavage principle, the method by which reproductive phasiRNAs targeted genes differed from that of most plant sRNAs, since other plant sRNAs usually target only one gene or several genes of the same gene family[38]. The degradome data here showed that reproductive phasiRNAs cleave their targets in a regulatory manner such that one phasiRNA can target more than one gene, and the target genes of one phasiRNA often belong to different gene families. This mechanism differs from that of plant microRNAs and tasiRNAs, which often target genes mainly within the same gene family[30,37,39]. This targeting approach might be more effective for gene silencing and increase error tolerance, such that the loss of function of one phasiRNA

might not induce serious defects during plant development. This phenomenon might also help to explain why the loss of MEL1 or DCL4 function led to complete sterility and meiosis perturbation, but that the knockdown of several miR2118 members only moderately negatively affected meiosis; moreover, the knockout of several phasiRNAs from a single *PHAS* locus only decreased the seed-setting rate in rice. Thus, these features may facilitate the role of phasiRNAs in mRNA elimination during early meiosis.

The third discovery of this study was the identification of the target genes of reproductive phasiRNAs, which are diverse but have major functions. Although piRNAs can also degrade various mRNAs in animals and in *C. elegans*, they appear to show no mRNA target specificity. However, 21-nt reproductive phasiRNAs mainly targeted rapidly downregulated genes that were enriched in adenyl ribonucleotide binding and serine/threonine kinase activity (especially RLKs). This contrasts with the targets of other plant siRNAs, indicating that these pathways are strictly regulated during meiosis in rice. Thus, the functions of reproductive phasiRNAs in rice might be involved in independent regulatory pathways, in addition to other sRNA regulatory pathways. Many studies have reported the roles of RLKs in early anther morphogenesis in plants, and these genes are often degraded when PMCs enter meiosis[22–26]. Very few studies have investigated the effects of the continuous expression of these genes on sporogenesis, but overexpression of LRR-RLK PK2 is detrimental to normal floral organ development[40], indicating that these RLKs are probably precisely regulated. The S-RLK proteins determine self-incompatibility responses in self-incompatible plants. In self-compatible plants, S-RLK expression is suppressed in the anthers, which might be a prerequisite for self-pollination, because the overexpression of S-RLKs in self-compatible plants often suppresses self-pollination[41]. We have also shown that high expression levels of the phasiRNA target genes negatively regulate sporogenesis and seed setting, and reproductive phasiRNAs are involved in suppressing these RLKs. These phasiRNAs also target genes that regulate gene transcription or RNA metabolism, which further eliminates mRNAs during early meiosis. The findings in this study suggest that the physiological function of the MEL1–phasiRNA complex might promote normal meiosis processes, such as eliminating specific mRNAs that are not required or that are harmful during meiosis and sporogenesis.

Finally, because phasiRNA functions depend on their nucleotide sequence, it is relevant to investigate how SNPs might have affected the functions of reproductive phasiRNAs during evolution, particularly because SNPs are essential for evolution and crop improvement. One important example is *PMS1T*, which generates 21-nt phasiRNAs that are associated with photoperiod-sensitive male sterility in rice[42]. A SNP in *PMS1T* near the miR2118-recognition site is critical for sporogenesis and probably leads to a differential accumulation of phasiRNAs[10]. In this study, we mimicked natural variation by mutating one or more nucleotides near the miR2118 cleavage sites in two independent reproductive phasiRNA precursor lncRNAs, to investigate how this affects the expression of phasiRNAs and their targets, as well as their roles in post-transcriptional regulation during meiosis in rice. We demonstrated that the sequence variation around the miR2118-binding site of the two chosen *PHAS* loci is essential for the precise expression of phasiRNAs and their targets, for subsequent pollen fertility and for the amount of seed set in rice. The sequences around miRNA binding site have also been proved to be important for the production of ta-siRNAs[43]. The results also show that nucleotide sequence integrity is important for phasiRNA function and pollen-grain development.

## Methods

**Plant growth conditions, generation of transgenic rice plants, and phenotype analysis**. The growth conditions and generation of transgenic plants were conducted according to Zhang et al.[44] The Zhonghua 11 (*Oryza sativa japonica*) rice cultivar was used for experiments. Rice plants were grown in the field in Guangzhou, China (23°08′ N, 113°18′ E), where the growing season extends from late April to late September. The mean minimum temperature range was 22.9–25.5 °C, and the mean maximum temperature range was 29.7–32.9 °C. The daylength ranged from 12 to 13.5 h. Plants were cultivated using routine management practices.

The *OsMEL1*, *MSPPL1*, and *MSPPL2* knockout mutants were generated using CRISPR-Cas9-based genome editing technology.[45] The $T_2$ and $T_3$ generations of the transgenic plants were used for phenotypic analyses. Heterozygous *mel1* plants were harvested and used to screen for homozygous *mel1* plants in the next generation, as homozygous *mel1* plants are completely sterile. The phenotypes in the $T_2$ and $T_3$ generations were stable. The primers are listed in Supplementary Data 6. The overexpression transgenic plants were constructed using PRHV-vector[46]. The *DCL4* RNAi mutant (*dcl4*) was obtained from Prof. Xiaofeng Cao (University of the Chinese Academy of Sciences)[47]. The miR2118j-g knockdown plant (*mir2118*) was constructed by Prof. Xuemei Chen (University of California, Riverside, California 92521, USA). The miR2118 family in rice consists of 18 mature miR2118 members with different sequences. The *mir2118* mutant was generated by mimicking the targets of miR2118j-g, which have similar sequences. An oligonucleotide containing the STTM-miR2118j-g sequence (5′-TAGGAA TGGGAGGCATCAGGAA-3′ and 5′-TAGGAATGGGAGGCATTAGGAA-3′) was chemically synthesized[48] and was inserted into the pCAMBIA1390 binary vector driven by the *UBI* promoter.

The phenotypes of anthers and pollen grains were analyzed at the heading stage. The phenotypes of plants, panicles, and rates of seed setting (the ratio of the number of filled grains to the total number of florets in a panicle) were determined when the seeds were harvested. For each line, data from 15 or more individual plants were obtained and statistically analyzed. Transgenic plants that were transformed with empty vector and WT plants were used as controls for the transgenic overexpressing plants. For CRISPR-Cas9 edited mutants, transgenic plants transformed with CRISPR-Cas9 vectors but that were not edited at the target sites, and WT plants were used as controls. Both these controls showed no obvious difference in both growth and seed-setting rate.

**Plant material collection, RNA extraction, and whole-transcriptome sequencing**. The spikelets from WT and *mel1* plants were collected at different developmental stages. Half of the spikelets were fixed and used for 4′,6-diamidino-2-phenylindole (DAPI) staining to analyze their developmental stage. The other spikelets were divided into four stages: PFS, PPS, PMDS, and EMS for WT, and the *mel1* spikelets were divided into the PFS and PPS stages. Total RNA was extracted with RNAiso plus Reagent (Takara, Japan) for three biological replicates for each sample and was used for sequencing. The preparation of whole-transcriptome libraries and deep sequencing were performed by the Annoroad Gene Technology Corporation. Libraries were controlled for quality and quantitated using the BioAnalyzer 2100 system and qPCR (Kapa Biosystems, Woburn, MA). The resulting libraries were sequenced initially on a HiSeq 2000 instrument that generated paired-end reads of 100 nt. The rice transcriptome was assembled using the Cufflinks 2.0 package according to the instructions provided[49]. Briefly, each RNA-

Seq dataset was aligned to the rice genome independently using the TopHat 2.0 program[50]. The transcriptome from each dataset was then assembled independently using the Cufflinks 2.0 program. All transcriptomes were pooled and merged to generate a final transcriptome using Cuffmerge. After the final transcriptome was produced, Cuffdiff was used to estimate the abundance of all transcripts based on the final transcriptome, and a BAM file was generated from the TopHat alignment.

**sRNA sequencing, PhasiRNA-generating loci (*PHAS*), and PhasiRNA trigger identification**. The *Oryza sativa* genome assembly RGAP 7.0 was used throughout this study and was downloaded from http://rice.plantbiology.msu.edu/[51]. Equal amounts of total RNA from three biological replicates of each of the five samples mentioned above were used for sRNA sequencing. The sRNA libraries were constructed and sequenced on an Illumina HiSeq 2000 platform at the Annoroad Gene Technology Corporation.

Genome-wide searching for *PHAS* loci and phasiRNA triggers, and phasiRNA expression analysis were performed using the PHASIS package (v3) according to the instructions (https://github.com/atulkakrana/PHASIS/wiki). Briefly, we used *phasdetect* for the de novo prediction of *PHAS* loci or precursor transcripts. Loci with a *p*-value < 0.001 were considered to be reliable *PHAS* loci. Phastrigs was then used to predict sRNA triggers for *PHAS* loci and precursor transcripts were identified using rice microRNAs, other reported rice sRNAs[52], and phasiRNAs. We also provided degradome data to experimentally support the sRNA triggers and to improve the confidence of triggers. The MEL1 RIP-data were obtained from publicly available sources (DDBJ: DRP000161[4]).

**Degradome sequencing and phasiRNA target identification**. The RNA amount and purity from the five samples were quantified using NanoDrop ND-2000 (NanoDrop, Wilmington, DE, USA). The RNA integrity was assessed by Agilent 2100 with RIN numbers > 7.0. Poly(A) RNA was purified from total RNA (20 µg) using poly-T oligo-attached magnetic beads using two rounds of purification. Because the 3′ cleavage product of the mRNA contains a 5′-monophosphate, the 5′ adapters were ligated to the 5′ end of the 3′ cleavage product of the mRNA by RNA ligase. First-strand cDNA synthesis was performed by reverse transcription with a 3′-adapter random primer, and size selection was performed with AMPureXP beads. The cDNA was amplified with PCR using the following conditions: initial denaturation at 95 °C for 3 min; 15 cycles of denaturation at 98 °C for 15 s, annealing at 60 °C for 15 s, and extension at 72 °C for 30 s; and then final extension at 72 °C for 5 min. The mean insert size for the final cDNA library was 200–400 bp. The 50-bp single-end sequencing was performed on an Illumina Hiseq 2500 (LC Bio, China) following the vendor's recommended protocol. Degradome sequencing was performed by the LC Bio Corporation. Genome-wide identification of phsiRNA targets was performed using the sPARTA package (v1.26)[53]. Putative targets were first predicted using the sPARTA built-in target prediction module with a standard scoring schema and a score cutoff of ≤5. For PARE validation, we used degradome sequencing data after sequences were cleaned of adapters and artifacts and were filtered for quality (Corrected *p*-value < 0.05, PARE read abundance ≥ 4 and PARE read abundance above the median for the gene).

**Polysome isolation and fractionation**. Total polysome isolation and fractionation were performed as described previously[54] with a little modification. Briefly,1 g WT PFS spikelets were ground in liquid nitrogen and resuspended in 8 mL polysome extraction buffer (0.2 M Tris-HCl, pH9.0, 0.2 M KCl,0.025 M EGTA, 0.035 M MgCl2, 1% Igepal CA630, 5 mM DTT, 1 mM PMSF, 100 µg/mL cycloheximide, 1% sodium deoxycholate, and 40 U/mL RNase inhibitor). The lysates were centrifuged at 13,000 g for 10 min at 4 °C. The supernatant was loaded onto an 8 mL 1.75 M sucrose cushion, and ribosomes were pelleted at 170,000 × g for 3 hr at 4 °C in a Type70 Ti rotor (Beckman Coulter) and resuspended in 400 µL resuspension buffer (0.2 M Tris-HCl, pH 9.0, 0.2 M KCl, 0.025 M EGTA, 0.035 M MgCl₂, 5 mM DTT, 100 µg/mL cycloheximide) for protein isolation and analysis of polysome profiles, and polysomes were separated by 10–45% sucrose density gradient centrifugation. 18 fractions were collected for each sample with continuous monitoring at 260 nm using a BioComp Piston Gradient Fractionator equipped with a Bio-Rad Econo UV Monitor.

**Rice protoplast transient transformation**. Two-week-old rice shoots were used for protoplast isolation. A bundle of rice plants (approximately 30 seedlings) was cut into approximately 0.5-mm strips with propulsive force using sharp razors. The strips were incubated in enzyme solution (1.5% cellulose RS, 0.75% macerozyme R-10, 0.6 M mannitol, 10 mM MES, pH 5.7, 10 mM CaCl₂, and 0.1% BSA) for 4–5 h in the dark with gentle shaking (40–50 rpm). Following enzymatic digestion, an equal volume of W5 solution (154 mM NaCl, 125 mM CaCl₂, 5 mM KCl, and 2 mM MES, pH 5.7) was added, and samples were shaken (60–80 rpm) for 30 min. Protoplasts were released by filtering through 40-µm nylon mesh into round-bottom tubes and were washed three to five times with W5 solution. The pellets were collected by centrifugation at 150 g for 5 min in a swinging bucket. After washing with W5 solution, the pellets were then resuspended in MMG solution (0.4 M mannitol, 15 mM MgCl₂ and 4 mM MES, pH 5.7) at a concentration of 2 × 10⁶ cells mL⁻¹. Aliquots of protoplasts (200 µL) were transferred into a 2-mL

round-bottom microcentrifuge tube and mixed gently with plasmid DNA (1 μg for phasiRNA target genes and 5 μg for *MEL1*) or with siRNA (5 μg). In control transfections, equivalent volumes of deionized, sterile water (mock-transfection) were added. Transfected protoplasts were collected by centrifugation for 5 min at $100 \times g$, resuspended and then were incubated at 28 °C in the dark for 20 h.

**Rapid amplification of 5′ cDNA ends (5′-RACE).** 5′-RLM-RACE was performed using the GeneRacer kit (Invitrogen) with three biological replicates. Without treating with Calf Intestine Alkaline (CIP), 1 μg total RNA was directly ligated to the RNA adapter oligonucleotide using T4 RNA ligase. After phenol/chloroform extraction and ethanol precipitation, reverse transcription of adapter ligated RNA was performed using random primers and SuperScript III™ RT. To confirm the site of mRNA cleavage, nested PCR was performed using the reverse gene-specific primer (Reverse GSP) and the GeneRacer™ 5′ Primer (homologous to the GeneRacer™ RNA oligonucleotide). The PCR products were purified and cloned using the p-EASY-blunt simple Cloning kit (TransGen), and the 5′-RACE clones were sequenced.

**qRT-PCR of phasiRNAs and their precursors and targets.** Total RNA from rice spikelets at the PFS, PPS, PMDS, and EMS were extracted with RNAiso plus Reagent (Takara, Japan) and reverse-transcribed using the PrimeScript™ RT reagent kit (Takara, Japan). RT-PCR was carried out using SYBR Premix Ex Taq™ (Takara, Japan). *U6* and *ACTIN2*, which were relatively stable in our sequencing data, were used as reference genes. The RT-PCR was performed according to the manufacturer's instructions (Takara, Japan), and the resulting melting curves were visually inspected to ensure the specificity of the product detection. Quantification of gene expression was performed using the comparative Ct method. Experiments were performed in triplicate, and the results are represented as the mean ± standard deviation (SD). The primers are listed in Supplementary Data 6.

**Reporting Summary.** Further information on research design is available in the Nature Research Reporting Summary linked to this article.

## Data availability
The sRNA-Seq, degradome, and transcriptome datasets were uploaded to the NCBI SRA database (SRA Accession No. PRJNA627552). The MEL1 RIP-data were obtained from publicly available sources (DDBJ: DRP000161[4]). Source data are provided with this paper.

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

## Acknowledgements

We thank Ms. Huang QJ for technique support. This research was supported by the National Natural Science Foundation of China (No. 91640202, 91940301, U1901202, 32070624, and 31770883), and grants from Guangdong Province (No. 2019JC05N394, 2017TQ04N779, 2019A1515011728), and grants from Sun Yat-Sen University (No. 20lgzd26).

## Author contributions

Y.C.Z. and M.Q.L. carried out the genome analysis and functional study and drafted the manuscript. Y.F.Z., Y.W.Y., J.P.L., Y.Y., Y.Z.F., R.R.H., H.H., and Z.Z. carried out mutant screening and functional experiments. K.R.Z. and J.H.Y. carried out phasiRNA identification. Y.Q.C. conceived the study, participated in its design and coordination, and helped to draft the manuscript. All authors read and approved the final manuscript.

## Competing interests

The authors declare no competing interests.
