## [Peer Review File · Nature Communications]

REVIEWER COMMENTS

Reviewer #1 (Remarks to the Author):

Zhang et al. reveal a so far unpublished function of rice 21nt-long reproductive phasiRNAs, namely targeting of specific mRNAs for elimination by cleavage, which in turn enables regular meiotic progression and male fertility. They found evidence for phasiRNA function in trans by studying mel1 (a rice AGO5 mutant which binds 21ntphasiRNAs) with sRNA-seq, RNA-seq and degradome sequencing. Importantly and astonishingly, functional follow-up characterization of mutants of phasiRNA loci and of overexpression of random target genes mostly resulted in obvious decrease in seed set, solidifying the wide importance for proper male development.

The manuscript is well written, precise, and convincing. Experimental approaches were well chosen to check for e.g. potential for translational inhibition (fractionation of lysate into cytosolic and polysomic fractions). The findings were also neatly put into context with other relevant literature from the animal and plant field. One outstanding question is whether the detected fertility decrease likely occurred high-temperature-dependent as reported for e.g. maize dcl5 (Teng et al. 2018, bioRxiv) and rice PMS1T lines. The manuscript will be of big interest in the field of small RNAs, particularly reproductive phasiRNAs.

The further comments and the ones in the edited pdf's are only suggestions and points to clarify, otherwise the manuscript is in a very good shape.

In Fig S2, 3,184 final target mRNAs were identified, from 135,969 potential target pairs (about 2%). It would be insightful to narrow down what makes a good potential target pair, e.g. regarding mispair score, PARE abundance, individual phasiRNA reads; MEL1-bound (Komiya et al 2014). Otherwise it seems a bit random.

In Fig S3a and Fig 1d, there is one replicate each that behaves differently (3rd PMC and 2nd EMS, respectively). Any possible explanation why?

As a note of caution, even random sets of sRNAs can produce patterns where one sRNA targets multiple genes, and one gene is hit by multiple sRNAs.

Minor points:

- Line 89f, stage names: PMS = PMC Meiosis Stage, set apart from PPS (PMC Prophase Stage) although prophase is part of meiosis; maybe PMS -> PMDS (PMC Meiotic Divisions Stage)?; also this contrast with Fig S2 where PPS is stated as "PMC premeiosis stage" – which is correct? Please check all occurrences and be consistent.
- FigS2: in addition, were panicles or spikelets sampled?
- Line 91: just for completeness, add the kind of mutation obtained to Fig S1 (frameshift/aa change...)
- Line 115: PARE read abundance – is there a normalization done? Or would using the sPARTA class schema be better?
- Line 147: how many percent of all target genes are kinases?
- Line 190: how were dcl4 and miR2118 knockdown created?
- Line 210 and around, and Fig5a: Please verify positions of miR2118 binding sites, PAMs and obtained mutations
- SRA upload: already done or not (I could not find it)
- Fig2a: small discrepancy: 70% DPKs instead of 68% as in text; also please clarify if the 100% in total from the divided circle are equal to the DPK or a subpart of them (then, how many percent?)
- In Fig 2b, there seems to be also a high number of genes upregulated in mel1 – any possible explanation or insight?
- Fig3c is too small and of insufficient quality to judge the result

Reviewer #2 (Remarks to the Author):

The work from Zhang and colleagues focus on the identification of targets of a group of phasiRNAs known for being specifically expressed in rice inflorescence and to be involved in gametogenesis.

The work also attempts to characterize the mode of action and molecular mechanisms of these sRNAs.

The first part of this work (mainly figures 1 and 2) is properly designed and executed, with conclusions well-based. Given the importance of rice and the involvement of this pathway with the plant fertility, the results presented here are of special interest. These findings have an additional relevance due to the fact that prior attempts have failed to find such targets (Song et al 2011, doi: 10.1111/j.1365-313X.2011.04805.x and Komiya et al 2014, doi: 10.1111/tpj.12483). However, I believe that the analysis of the targets and the phasiRNAs involved could be expanded. For instance:

- How many of the phasiRNAs had targets? Among the phasiRNAs with putative targets, from how many PHAS loci are they coming from? Is there one PHAS gene with more relevant contribution? Is there any phasiRNA MEL1-independent and maybe loaded in another AGO?

Unfortunately, the second part of the manuscript is quite confusing, with conclusions that are not well supported and also seem to be redundant with the literature, and even ignore some of the current knowledge about phasiRNA biology (for recent reviews on phasiRNAs, please refer to these recent reviews – de Felippes 2019 in Plants and Deng et al 2018 in Plant biotech Journal).

Given the interesting findings of the first part, I would suggest the authors to focus on this part, including some expansion as suggested above. There are some experiments on the second part though, which in my opinion are quite welcomed. For example, given that most of the characterization of new PHAS loci is mainly done bioinformatically, some “wet-biology” confirmation is really appreciated, as the mutations of the miR2118 target site to confirm the importance of this miRNA in triggering secondary siRNAs. Nonetheless, most of these experiments would need to be re-designed.

Bellow are a few more comments/suggestions that I believe could improve the manuscript:

- A section in introduction about phasiRNAs, including the role of miR2118 would be definitely helpful for the reader
- It could be interesting to explore and discuss the dependency of these phasiRNAs to MEL1. Based on other PHAS systems, I would expect miR2118 to trigger phasiRNA production through AGO1, with the resulting phasiRNAs being loaded in MEL1 after being produced. Komiya et al 2016 has shown that miR2118 is also loaded in MEL1, but the requirement of MEL1 for these phasiRNA production was not clear.
- In figure 1c, it would be nice to have a positive control, i.e. a miRNA that is known to act mainly through translation inhibition and to associate with polysomes (see Brodersen 2008).
- When presenting the number of targets, the way the number is written (with commas) is confusing (for ex line 118).
- Line 121, how were the 3184 targets selected? Did they had 2x fold variation? Etc..
- Figure 1e, Confirmation by RACE would be welcomed.
- In figure 2a and 3, the following expression “MD-phasiRNA-target kinases” is not clear.
- Is MSP1 also targeted by phasiRNAs?
- When overexpressing some of the targets (figure 2c-e) one would expected these to be also silenced by the endogenous phasiRNAs, no?
- The analysis of chromosomal preference (fig 3a) seems unnecessary. Small RNAs, included phasiRNAs rely on sequence specificity, in this sense, chromosomal location is irrelevant. It would be more interesting to know if genes targeted by the same phasiRNAs are related.
- Figure 3b is very imprecise. It suggests that the 21nt long phasiRNAs or MEL1-loaded siRNAs are triggering the production of secondary siRNAs. There is no data supporting this claim.
- The authors claim to have found a new mode of action for phasiRNAs in plants, with one sRNA targeting several genes, different from miRNAs. There are several miRNAs that target several different genes (miR173, 156, 159, etc). Moreover, phasiRNAs targeting several related genes is a

known process, and one of the main characteristic of these molecules (Fei 2013, Plant Cell).
- Experiments in figure 5/6. As mentioned before, I do appreciate this kind of "wet-lab" experiments. However, it is not clear which were the modifications made, where it was made and with which purpose. Also the conclusions seem redundant. Given that sRNAs rely on sequence, it is more than obvious that mutations changing sequence could affect targeting, not to mention that change in phasing.

Reviewer #3 (Remarks to the Author):

This manuscript attempts to test the hypothesis that the vast number of reproductive phasiRNAs found during rice pollen development modulate mRNA abundance and are essential for pollen maturation. While the manuscript includes some interesting approaches, including making PHAS locus edits and over-expressing phasiRNA targets, many experiments are improperly controlled or poorly described, and others appear to apply circular logic. My concerns are detailed below, but chief among them are:

1. If expression changes were used to help narrow the list of putative phasiRNA targets, expression changes cannot be a conclusion from any of the subsequent analysis.
2. Experiments are poorly controlled. Transgenic experiments should be controlled with wild-type siblings. (To be fair, this might be what was done, but it is not mentioned and it therefore appears to be true WT.) Additional transgenic controls are also needed to be assured that disruption of phasiRNA regulation causes adverse phenotypes, rather than just misexpression of developmental regulators. Crosses are needed to demonstrate that seed set defects are caused by the pollen genotype.
3. Experiments and analysis are poorly explained. There is no information about specific alleles created through genome editing, nor extent of over-expression for transgenics. There is no description of the relationship between PHAS loci, phasiRNAs and targets that are analysed in depth. It is also not explained how data from ref. 1 was re-analyzed to draw conclusions here.

The authors have put tremendous effort into creating transgenic material and sequencing datasets that could be very informative. I encourage them to follow through with the careful experimentation and writing that these materials deserve.

Figure S1: Please include more information on the specific allele of mel1 created. What is the change at the target site? Also, please indicate which part of the protein is detected by the antibody in panel d. The reader shouldn't have to dig into a reference for this critical information.

I'm concerned about rigour in PARE validation of predicted phasiRNA targets. The manuscript reports that p-values were <0.05 , but does not report whether these values have been FDR corrected. (I assume this is the outputted p-value from sPARTA, but does that program implement FDR?) The sPARTA manuscript (Kakrana et al) is very clear that its method for determining p-values is not highly stringent and "have a higher proportion of false positives". This likely explains the high number of predicted targets in the manuscript. Kakrana suggest including replicates or using expression to narrow the list of targets. It appears that the latter was used in this manuscript, although there is no explanation in lines 118-120 as to how.

I assume that evidence of down-regulation was used (implied in Figure S2). But is this based on developmental dynamics or impact of the mutant? Either way, if down-regulation is a criterion for

classification as a phasiRNA target, then Figure 1d is biased. You cannot use expression as a criterion for selection and then report expression changes as a characteristic of the selected genes. (Also the legend for this panel is insufficient. Is this a log scale? What are the values compared against?)

Even with this bias toward predicted targets with the expected expression pattern, I do not think Figure 1d supports the statement that "the expression was gradually down regulated with meiosis progression". There is a clear shift from PFS to PPS, but changes after that point are very small. The huge variation in EMS replicates calls into question whether such small changes are meaningful.

I am also concerned about potential bias in Figure 1e. It is not clear to me how cleavage position factors into SPARTA's prediction of target genes. Are PARE reads showing cleavage at a non-standard position weighed the same as PARE reads with expected cleavage? If not, then Figure 1e is biased in the same way as Figure 1d.

Lines 135-144: This section is confusing. Because the citations are numerical, I have to look at the reference list to even figure out which paper contains this "data from maize". It would be much better to explain in the main text what this data is and how it was analyzed, particularly as the cited reference concludes that "clustering removed information about developmental dynamics" and they abandoned this method. Therefore when this paper attempts to overlap GO enrichment terms between "reported cluster 1" and their rice phasiRNA targets, I have no idea why that is valid. To be fair, I did not dig through the supplemental materials for the cited paper, but should I have to in order to understand this one?

The conclusion of this section of results is that "MEL1 is indispensable for reprogramming of mRNA expression during early meiosis." If I understand correctly, this conclusion is based on overlap between GO terms enriched among phasiRNA targets and genes upregulated in mel1, or genes shown to vary during maize development. I am again concerned that gene expression changes were used to *define* the phasiRNA targets, making this logic circular.

Figure 2c-e: for the transgenic lines, wild-type is not a sufficient control. Over-expression of a non-phasiRNA targeted gene (but critically, a gene expressed during pollen development!!) is required. This experiment just shows that genes selected for pollen expression (and possibly selected for dynamic pollen expression) need to be correctly expressed for normal pollen development. (Or even worse, it shows that the transformation process impacts fertility.) An even better experiment would be to express these genes by their native promoter but with silent mutations that eliminate phasiRNA binding.

I have a similar concern for experiments shown in Fig 5b-c and Fig 6. The appropriate control for these edited plants are heterozygous or homozygous WT siblings from a segregating population. Otherwise there is no control for the effects of transgenesis or other potential changes in the background.

Figure 3C is an important panel to verify the PARE data, which otherwise has no replication. Please explain this panel better and make fonts larger to be more legible. Is "seq-40774" a phasiRNA? Does the arrow with "5/7" indicate 5 of 7 cleavage products were found at that site? Also how were these three targets selected? Was RLM-RACE attempted with additional targets, or were these the only successful examples?

The description of a mi2118 mutant in rice is interesting, however as this line was provided by another laboratory, has it been published elsewhere? If not, please provide full details of how it was created (sequence of STTM, for example). Also, how was miR2118 expression quantified in Figure S5? The methods include qRT-PCR of phasiRNAs, but not miRNAs.

In Figure 4b, please report how many pollen grains were assessed at each anther length, how many anthers per size class, and how many plants those anthers were derived from. The methods state that at least 30 plants were analyzed, but it's not clear which analysis this refers to. As an aesthetic point, I find this type of 3D bar graph very difficult to interpret due to bars overlapping, perspective distorting sizes, etc.

In Figure 4C it is not possible to see the red arrows that indicate chromosomal abnormalities, nor are these images of sufficient quality to identify those changes myself.

Seeing some inviable pollen (Fig 6a) is not sufficient to determine that "decreased fertility is caused by aborted pollen grains". Plants make far more pollen than they need and it is common to have pollen defects with no impact on fertility. The correct experiment is reciprocal crosses between *msppl* mutants and WT to demonstrate that seed set is controlled by paternal genotype.

The conclusions in lines 226-229 are not supported by the data. I cannot conclude that "sequences downstream from the miR2118-binding site are more important than those upstream" when there are no direct comparisons between these - only comparisons with WT. Even if that comparison were made, it would only apply to the PHAS locus for which there is data. More examples are needed to make a generalization about all PHAS loci.

It is difficult to interpret the data in Fig 6b-e without a schematic of the PHAS locus that shows where these phasiRNA sequences are generated in relation to the miR2118 site and the edits. Are the two "edited" phasiRNAs in Fig6b experimentally confirmed through sequencing? If not, how was their abundance assessed? The legend says that these are "edited" in the *msppl1* mutant, but at least three different alleles for *msppl1* are described - are they predicted to be created in all alleles?

Figure 6c show *msppl1-1* and *msppl1-2*, which I gather are the two independent lines of *msppl1* described in lines 243-244. Both are "PAM2-edited", but does that mean they are siblings with the same allele, or does it mean they are independent edits from the PAM2 transgenic line? Throughout this section, I have a lot of confusion about the different alleles created by these edits and would like to see the full genotypes of all plants.

Figure 6 needs statistical testing and confirmation that the replicates were independent biological replicates (ideally, different plants) and not technical replicates.

The data in Figure S5b don't make sense to me. Isn't deletion of the miR2118 expected to cause an increase in the PHAS transcript by eliminating its conversion to phasiRNAs?

For dataset 1, please provide the number of mapped reads in addition to total reads.

Figure S3a needs more description - what is the scale, how were samples normalized?

In Figure 1f I assume "reproductive" phasiRNAs were those identified in this study. What are the "other" phasiRNAs?

Figure 3A is not a useful panel in the manuscript. And Figure 3b is a model that is more meaningful as a final figure.

How was "seed setting rate" determined (Fig 5c)? Is this based on total seed number? Seed weight? Rate of plants that set any seed at all?

Point-by-point response to reviewers' comments and questions

Reviewer Comments:

Reviewer #1:

Comments:

Zhang et al. reveal a so far unpublished function of rice 21nt-long reproductive phasiRNAs, namely targeting of specific mRNAs for elimination by cleavage, which in turn enables regular meiotic progression and male fertility. They found evidence for phasiRNA function in trans by studying *me11* (a rice AGO5 mutant which binds 21ntphasiRNAs) with sRNA-seq, RNA-seq and degradome sequencing. Importantly and astonishingly, functional follow-up characterization of mutants of phasiRNA loci and of overexpression of random target genes mostly resulted in obvious decrease in seed set, solidifying the wide importance for proper male development.

The manuscript is well written, precise, and convincing. Experimental approaches were well chosen to check for e.g. potential for translational inhibition (fractionation of lysate into cytosolic and polysomic fractions). The findings were also neatly put into context with other relevant literature from the animal and plant field.

Comment 1. One outstanding question is whether the detected fertility decrease likely occurred high-temperature-dependent as reported for e.g. maize *dcl5* (Teng et al. 2018, bioRxiv) and rice PMS1T lines. The manuscript will be of big interest in the field of small RNAs, particularly reproductive phasiRNAs. The further comments and the ones in the edited pdf's are only suggestions and points to clarify, otherwise the manuscript is in a very good shape.

Reply: Thank you for the comments. Following the suggestion, we have compared the seed setting rates of a certain transgenic lines which were grown from March to July in Guangzhou, China (the mean maximum temperature during heading is ~33°C)

or from July to November in Guangzhou, China (the mean maximum temperature during heading is ~30°C). The seed setting rate variation is showed in the **Figure 1 below**. We did not found significant difference between the seed setting rate of the transgenic plants growing in the two seasons. But this result cannot exclude whether fertility decreases in a higher temperature.

Figure 1. Seed setting rate of mutants grown under high-temperature (HT) and low-temperature (LT) conditions. The values are the means \pm SDs ($n > 15$ plants for each transgenic line)

Comment 2. In Fig S2, 3,184 final target mRNAs were identified, from 135,969 potential target pairs (about 2%). It would be insightful to narrow down what makes a good potential target pair, e.g. regarding mispair score, PARE abundance, individual phasiRNA reads; MEL1-bound (Komiya et al 2014). Otherwise it seems a bit random.

Reply: The original 135,969 potential target pairs included potential complementary sites of all the sequenced phasiRNAs on both mRNA and TE. We agree with the comment and have narrowed down the predicted target pairs by using reproductive phasiRNAs (highly expressed in PFS and significantly decreased at EMS) to predict the target pairs on mRNAs (mispair score ≤ 5) in the revised manuscript, and 32,993

potential target pairs are obtained. We have also narrowed down the PARE data validated target pairs by using corrected P-value (≤ 0.05) instead of P-value (≤ 0.05), and 3,437 target pairs are identified (**data S3**). We have revised the manuscript accordingly (**page 6, line 120, marked in blue**). Thank you for the suggestion.

Comment 3. In Fig S3a and Fig 1d, there is one replicate each that behaves differently (3rd PMC and 2nd EMS, respectively). Any possible explanation why?

Reply: Yes, we saw the difference. The differences might be due to the experimental bias during RNA extraction or library construction. To reduce the experimental bias, we have used sRNA-seq data of WT PFS and EMS to identify reproductive phasiRNAs, and used transcriptome data of mel1 PPS, WT PFS and PPS to filter targets, this analysis method could reduce the influence of the one sample variation (sRNA-seq of 3rd PMS and transcriptome of 2nd EMS). We appreciate your comment.

Comment 4. As a note of caution, even random sets of sRNAs can produce patterns where one sRNA targets multiple genes, and one gene is hit by multiple sRNAs.

Reply: Thank you for the indication. We have rephrased the corresponding sentences in the revised manuscript (**page 15, line 327-331, marked in blue**).

Minor points:

Comment 5.- Line 89f, stage names: PMS = PMC Meiosis Stage, set apart from PPS (PMC Prophase Stage) although prophase is part of meiosis; maybe PMS -> PMDS (PMC Meiotic Divisions Stage)?; also this contrast with Fig S2 where PPS is stated as “PMC premeiosis stage” – which is correct? Please check all occurrences and be consistent.

Reply: Thank you for the suggestion. We have renamed PMS to PMDS as suggested, and PPS to PMC (Prophase Stage) in the revised manuscript.

Comment 6. Fig S2: in addition, were panicles or spikelets sampled?

Reply: Spikelets were sampled for the sequencing. We have revised Figure S2, thank you.

Comment 7. Line 91: just for completeness, add the kind of mutation obtained to Fig S1 (frameshift/aa change...)

Reply: The mutation sites have been added to **Fig S1a** as suggested, thank you.

Comment 8. Line 115: PARE read abundance – is there a normalization done? Or would using the sPARTA class schema be better?

Reply: The PARE read abundance has been normalized. We have also added the sPARTA class information to the revised **data S3** as suggested, thank you.

Comment 9. Line 147: how many percent of all target genes are kinases?

Reply: 11.7% of the identified target genes which have GO annotations belong to kinases family. The information has been added to the revised manuscript accordingly (**page 7, line 154, marked in blue**).

Comment 10. Line 190: how were *dcl4* and *mir2118* knockdown created?

Reply: *dcl4* mutant has been constructed by RNAi technology, and *mir2118* knockdown mutant has been constructed by STTM technology. According to the comment, we have added the detailed information into the revised methods section as suggested (**page 19, line 402-409, marked in blue**).

Comment 11. Line 210 and around, and Fig5a: Please verify positions of miR2118 binding sites, PAMs and obtained mutations

Reply: We have provided a figure (**Fig S7b**) and added the positions of potential miR2118 binding sites, PAMs and the mutation sequence accordingly. Indeed, this information can make readers more easily to read, thank you.

Comment 12. SRA upload: already done or not (I could not find it)

Reply: We have uploaded all the datasets to SRA database (PRJNA627552), and the data will be released soon.

Comment 13. Fig2a: small discrepancy: 70% DPKs instead of 68% as in text; also please clarify if the 100% in total from the divided circle are equal to the DPK or a subpart of them (then, how many percent?)

Reply: Thank you for the indication. We have revised the text to 71.2% in the revised manuscript (**page 8, line 155, marked in blue**). **Fig 2a** was also revised to make it readable

Comment 14. In Fig 2b, there seems to be also a high number of genes upregulated in *mell* – any possible explanation or insight?

Reply: This is a good comment indeed. Our result showed that there are more genes which were upregulated in *mell* plants than genes that downregulated in *mell* plants (**Fig 2b**). One probable reason is the failure of RNA cleave mediated by MEL1, another reason might be the cascade effect that the phasiRNA targets include potential signaling proteins and transcriptional factors. Also, a recent study about piRNAs in mammals reported that some piRNAs can activate gene expression¹. We cannot exclude the possibility that some phasiRNAs activate gene expression just like

piRNAs in mammals, this deserves to further investigation.

Comment 15. Fig3c is too small and of insufficient quality to judge the result

Reply: We have revised the original Fig 3c (which has been changed as **Fig 3b** in the revised manuscript) to make it clear. In addition, according to the comment 8 from reviewer 2, we have performed additional RACE validations of the cleavage of phasiRNAs on their target genes as shown in the revised **Fig 3b**.

Reviewer #2 (Remarks to the Author):

The work from Zhang and colleagues focus on the identification of targets of a group of phasiRNAs known for being specifically expressed in rice inflorescence and to be involved in gametogenesis. The work also attempts to characterize the mode of action and molecular mechanisms of these sRNAs.

The first part of this work (mainly figures 1 and 2) is properly designed and executed, with conclusions well-based. Given the importance of rice and the involvement of this pathway with the plant fertility, the results presented here are of special interest. These findings have an additional relevance due to the fact that prior attempts have failed to find such targets (Song et al 2011, doi: 10.1111/j.1365-313X.2011.04805.x and Komiya et al 2014, doi: 10.1111/tpj.12483). However, I believe that the analysis of the targets and the phasiRNAs involved could be expanded. For instance:

Comment 1. How many of the phasiRNAs had targets? Among the phasiRNAs with putative targets, from how many PHAS loci are they coming from? Is there one PHAS gene with more relevant contribution? Is there any phasiRNA MEL1-independent and maybe loaded in another AGO?

Reply: Thank you for the comments. 2,717 phasiRNAs (24.3% of total identified reproductive phasiRNAs) were found to have targets and they come from 1,408 PHAS loci. We have provided these data in the revised manuscript accordingly (**page 6, line 132-133, marked in blue**).

Regarding to the comment “is there one PHAS gene with more relevant contribution”, we analyzed the PHAS loci, the PHAS locus which has the most targets is 21nt:Phas-1866, but it only has 15 targets pairs, thus we cannot conclude if there is any one PHAS gene with more relevant contribution at this stage.

For the AGO loading capacity of phasiRNAs, it has been reported that several AGO proteins are expressed during rice reproductive development in addition to MEL1, such as AGO1b, 1c, 1d and AGO18^{2,3}. Moreover, knockdown of AGO18 reduced the accumulation of secondary siRNAs including those triggered by miR2118 in panicle. It is possible that phasiRNAs might also be loaded in other AGOs, but these speculations need to validate experimentally and bioinformatically.

Comment 2. Unfortunately, the second part of the manuscript is quite confusing, with conclusions that are not well supported and also seem to be redundant with the literature, and even ignore some of the current knowledge about phasiRNA biology (for recent reviews on phasiRNAs, please refer to these recent reviews – de Felippes 2019 in Plants and Deng et al 2018 in Plant biotech Journal). Given the interesting findings of the first part, I would suggest the authors to focus on this part, including some expansion as suggested above. There are some experiments on the second part though, which in my opinion are quite welcomed. For example, given that most of the characterization of new PHAS loci is mainly done bioinformatically, some “wet-biology” confirmation is really appreciated, as the mutations of the miR2118 target site to confirm the importance of this miRNA in triggering secondary siRNAs. Nonetheless, most of these experiments would need to be re-designed.

Reply: We understand the concern. According to the comment, we have performed

additional experiments and revised the manuscript as suggested, including 1) added positive controls of polysome analysis experiment (**Fig 1c**); 2) RACE experiments of more cleavage sites (**Fig 3b**) expression analysis of phasiRNA targets in transgenic plants (**Fig S5a**); 4) revised descriptions and figures such as the introduction of phaiRNAs, description of target screening and experiment detail, deletion of redundant sentences and citation of more appropriate references. For detail please see below the reply of each comment. Thank you these comments to improve the manuscript.

Comment 3. Bellow are a few more comments/suggestions that I believe could improve the manuscript:

- A section in introduction about phasiRNAs, including the role of miR2118 would be definitely helpful for the reader

Reply: We have added the introduction about phasiRNAs, including the role of miR2118 in the introduction section in the revised manuscript as suggested (**page 3, line 56-58, marked in blue**), thank you.

Comment 4. It could be interesting to explore and discuss the dependency of these phasiRNAs to MEL1. Based on other PHAS systems, I would expect miR2118 to trigger phasiRNA production through AGO1, with the resulting phasiRNAs being loaded in MEL1 after being produced. Komiya et al 2016 has shown that miR2118 is also loaded in MEL1, but the requirement of MEL1 for these phasiRNA production was not clear.

Reply: This is a good question indeed. We have interestingly found that knockout of MEL1 lead to down regulation of phasiRNAs especially 21nt phasiRNAs, and no apparent difference between phasiRNAs that triggered by miR2118 and those triggered by other 21nt phasiRNAs (please see below the figure 2). This observation is similar with that reported in AGO18 knockdown mutant, although the mechanism is

unclear². In addition to the requirement of these AGO proteins for phasiRNA biogenesis, these AGO proteins might be also involved in maintaining phasiRNAs that loaded to the AGO-phasiRNA complex.

Figure 2. Heatmap of reproductive phasiRNAs

Comment 5. In figure 1c, it would be nice to have a positive control, i.e. a miRNA that is known to act mainly through translation inhibition and to associate with polysomes (see Brodersen 2008).

Reply: Following your suggestion, we have added GAPDH and miR171 as the positive controls for further experimental validation as suggested. miR171 has been reported to regulate mRNA translation⁴. We showed that both miR171 and GAPDH indeed existed in polysome fractions (**Fig 1c**). As plant miRNAs mediate both translation inhibition and RNA cleavage, we also found that miR171 preferred to exist in supernatant fraction (**Fig 1c**), which is similar with that previously reported^{4,5}. miR171 showed a higher enrichment in polysome fractions than that of phasiRNAs. The data has been added to the revised manuscript (**page 5, line 107-109; page 6, line 111-112, marked in blue**) and Fig 1c. Thank you for the suggestion.

Comment 6. When presenting the number of targets, the way the number is written (with commas) is confusing (for ex line 118).

Reply: We have revised all the numbers using thousand separators in the revised manuscript, thank you.

Comment 7. Line 121, how were the 3184 targets selected? Did they had 2x fold variation? Etc..

Reply: The target selects are based on following principles: we firstly identified cleaved targets by combining target prediction and PARE sequencing target validation, we then filtered the targets by transcriptome data. Reproductive phasiRNAs have been reported to be load into MEL1 (at least partially) to perform function, and the abundance of mRNAs should not be higher in samples in which they were cleaved than in samples in which they were not cleaved. Thus the targets have more than 1.2x down regulation in *mell* sample than WT sample but were not cleaved in *mell* sample were then filtered out. The rest of the predicted targets are considered as the targets of reproductive phasiRNAs. According to the comment, we have revised the manuscript and added the description to make the statement more clear (**page 6, line 126-131, marked in blue**), thank you.

Comment 8. Figure 1e, Confirmation by RACE would be welcomed.

Reply: We agree and have performed more RACE experiments to verify the cleavage sites as suggested. The cleavage at 9th, 10th and 11th could be detected by RACE experiment in 10 genes of 15 chosen cleavage sites. The data have been added to the revised **Fig 3b**, thank you.

Comment 9. In figure 2a and 3, the following expression “MD-phasiRNA-target kinases” is not clear.

Reply: We have rephrased the description to “kinases targeted by reproductive

phasiRNAs ”, thank you.

Comment 10. Is MSP1 also targeted by phasiRNAs?

Reply: Yes, MSP1 is also targeted by two phasiRNAs, these data have been shown in **data S3**.

Comment 11. When overexpressing some of the targets (figure 2c-e) one would expect these to be also silenced by the endogenous phasiRNAs, no?

Reply: To answer this question, we have performed additional experiments to analyze the abundance of these target genes in both seedlings and panicles in transgenic plants. As shown in **Fig S5a**, the overexpression fold change of mRNA abundance of six out of 10 target genes were smaller in anthers than that in seedlings compared with WT plants, indicating that they are silenced by the endogenous phasiRNAs, but cannot be completely silenced, the situation is similar with those cases when overexpressing miRNA targets, the endogenous miRNAs also cannot completely silence the overexpressing targets. We appreciate the comment and the new data has been added to the revised **Fig S5a**, Thank you.

Comment 12. The analysis of chromosomal preference (fig 3a) seems unnecessary. Small RNAs, included phasiRNAs rely on sequence specificity, in this sense, chromosomal location is irrelevant. It would be more interesting to know if genes targeted by the same phasiRNAs are related.

Reply: We agree and have moved Fig 3a to supplementary Figures as **Fig S6c**, and deleted some descriptions in the revised manuscript as suggested. We also follow the suggestion to analyze if genes targeted by the same phasiRNAs are related, and found that genes targeted by the same phasiRNAs do not tend to originate from a same gene family, this new data has been provided in the revised manuscript (**page 9, line**

191-192, marked in blue). Thank you.

Comment 13. Figure 3b is very imprecise. It suggests that the 21nt long phasiRNAs or MEL1-loaded siRNAs are triggering the production of secondary siRNAs. There is no data supporting this claim.

Reply: According to the comment, we have revised Fig 3b to make the working model more accurate. We have also taken the suggestion from the reviewer 3 (comment 24) and moved Fig 3b to **Fig 6d**. We thank you for the suggestion.

Comment 14. The authors claim to have found a new mode of action for phasiRNAs in plants, with one sRNA targeting several genes, different from miRNAs. There are several miRNAs that target several different genes (miR173, 156, 159, etc). Moreover, phasiRNAs targeting several related genes is a known process, and one of the main characteristic of these molecules (Fei 2013, Plant Cell).

Reply: We agree and have rephrased the related sentences and cited the references mentioned above in the revised manuscript as suggested (**page 15, line 327-331, marked in blue**), thank you.

Comment 15. Experiments in figure 5/6. As mentioned before, I do appreciate this kind of “wet-lab” experiments. However, it is not clear which were the modifications made, where it was made and with which purpose. Also the conclusions seem redundant. Given that sRNAs rely on sequence, it is more than obvious that mutations changing sequence could affect targeting, not to mention that change in phasing.

Reply: We apologize not clearly showing the information in original version of the manuscript. According to the comments, we have provided a figure (**Fig S7b**) and added detailed information of the mutant including the genotypes of all the transgenic lines, and rephrased the related sentences referred to experiments in figure 5 and 6.

The experiment detail and purpose were added in the revised manuscript (**page 11, line 228-230; page 13, line 268-269, marked in blue**). We also deleted redundant conclusions in original version of the manuscript have been deleted (**page 12, line 258-261, marked in blue**). We have also moved Fig 6b, c to the supplementary figures (revised **Fig S8**) according to the comment, thank you for all your suggestion.

Reviewer #3 (Remarks to the Author):

This manuscript attempts to test the hypothesis that the vast number of reproductive phasiRNAs found during rice pollen development modulate mRNA abundance and are essential for pollen maturation. While the manuscript includes some interesting approaches, including making PHAS locus edits and over-expressing phasiRNA targets, many experiments are improperly controlled or poorly described, and others appear to apply circular logic. My concerns are detailed below, but chief among them are:

Comment 1. If expression changes were used to help narrow the list of putative phasiRNA targets, expression changes cannot be a conclusion from any of the subsequent analysis.

Reply: We agree with the comment. We used the expression changes to help narrow the list of putative targets. According to the comment, we have added the description of the target screening in the revised version of the manuscript and rephrased the description accordingly to make the statement clear (**page 6, line 126-131, marked in blue**), thank you for the indication.

Comment 2. Experiments are poorly controlled. Transgenic experiments should be controlled with wild-type siblings. (To be fair, this might be what was done, but it is not mentioned and it therefore appears to be true WT.) Additional transgenic controls are also needed to be assured that disruption of phasiRNA regulation causes adverse

phenotypes, rather than just misexpression of developmental regulators. Crosses are needed to demonstrate that seed set defects are caused by the pollen genotype.

Reply: We agree with the comment that additional transgenic controls are important to address the question. So we have used different transgenic plants as controls: for those transgenic plants overexpressing phasiRNA targets, transgenic plants that transfected with empty vector and wild type plants as the control. For the CRISPR-Cas9 edited mutants, we have used both transgenic plants that transfected with CRISPR-Cas9 vectors but have no edition in the target sites and wild type plants as the controls. We have also obtained the overexpression transgenic plants of non-phasiRNA targeted genes, including the genes that express in anther and genes that do not express in anther non-phasiRNA targeted gene (please also see the reply to comment 8 below). We observed that these control plants did not show obvious differences in growth and seed setting rate. The seed setting rate of all the control plants are shown below Figure 3 and revised **Fig S5b-e**. These additional transgenic controls included indeed help to further support that disruption of phasiRNA regulation causes adverse phenotypes. We have revised the methods section of the manuscript to make it clear (**page 19, line 414-419, marked in blue**). Thank you for the suggestion to improve the study.

According to the suggestion that to demonstrate that seed set defects are caused by the pollen genotype in phasiRNA target overexpression plants, *msppl1*, *msppl2*, and *mir2118* plants, we crossed these plants with WT plants as suggested. The results showed that when using these mutants as male parent, the seed setting rates significantly decreased compared with that using WT as male parent, indicating that the defects of pollen grains in these mutants impaired pollination and seed setting. The result has been added to the revised **Fig S6a, b** and manuscript (**page 9, line 183-188; page 11, line 237-238, marked in blue**), thank you.

Figure 3. Seed setting rates of different controls.

Comment 3. Experiments and analysis are poorly explained. There is no information about specific alleles created through genome editing, nor extent of over-expression for transgenics. There is no description of the relationship between PHAS loci, phasiRNAs and targets that are analysed in depth. It is also not explained how data from ref. 1 was re-analyzed to draw conclusions here.

Reply: We apology for not clearly explaining the experiments and analysis in previous version of the manuscript. Following your criticisms, we have provided detailed information about the alleles created through genome editing (**Fig S1a**) and the extent of overexpression for transgenic plants (**Fig S5a**), and provided a figure (**Fig S7b**) to add description of the PHAS loci, phasiRNAs and targets in the revised manuscript, thank you.

For the data of ref. 1, we have compared the significantly enriched F GO terms of phasiRNA targets with those previously reported enriched in differentially expressed genes during maize meiosis. According to the comments, we have explained more in the revised manuscript (**page 7, line 143-145, marked in blue**) and have labeled the overlapping F GO terms in the revised **Data S4** as suggested, thank you.

Comment 4. The authors have put tremendous effort into creating transgenic material

and sequencing datasets that could be very informative. I encourage them to follow through with the careful experimentation and writing that these materials deserve. Figure S1: Please include more information on the specific allele of mel1 created. What is the change at the target site? Also, please indicate which part of the protein is detected by the antibody in panel d. The reader shouldn't have to dig into a reference for this critical information.

Reply: Thank you for the comments and appreciate very much your encouragement. As suggested, we have provided more information of the allele of mel1 created as suggested (**Fig S1a**) in the revised manuscript. The MEL1 antibody detecting part was also showed in the revised **Fig S1c**, thank you.

Comment 5. I'm concerned about rigour in PARE validation of predicted phasiRNA targets. The manuscript reports that p-values were <0.05 , but does not report whether these values have been FDR corrected. (I assume this is the outputted p-value from sPARTA, but does that program implement FDR?) The sPARTA manuscript (Kakrana et al) is very clear that its method for determining p-values is not highly stringent and "have a higher proportion of false positives". This likely explains the high number of predicted targets in the manuscript. Kakrana suggest including replicates or using expression to narrow the list of targets. It appears that the latter was used in this manuscript, although there is no explanation in lines 118-120 as to how.

I assume that evidence of down-regulation was used (implied in Figure S2). But is this based on developmental dynamics or impact of the mutant? Either way, if down-regulation is a criterion for classification as a phasiRNA target, then Figure 1d is biased. You cannot use expression as a criterion for selection and then report expression changes as a characteristic of the selected genes. (Also the legend for this panel is insufficient. Is this is a log scale? What are the values compared against?)

Even with this bias toward predicted targets with the expected expression pattern, I do not think Figure 1d supports the statement that "the expression was gradually down regulated with meiosis progression". There is a clear shift from PFS to PPS, but

changes after that point are very small. The huge variation in EMS replicates calls into question whether such small changes are meaningful.

Reply: We thank you for the comment. In response to your comment, we have filtered the targets by corrected p-values in the revised manuscript as suggested, and the target gene number were cut down from 3,184 to 2,783, thank you.

In addition, we have also provided more information on the the target selection in the revised manuscript (**page 6, line 126-131, marked in blue**). Following principles were used: we identified cleaved targets by combining target prediction and PARE sequencing target validation, we then filtered the targets by transcriptome data. Reproductive phasiRNAs have been reported to be load into MEL1 (at least partially) to perform function, and the abundance of mRNAs should not be higher in samples in which they were cleaved than in samples in which they were not cleaved. Thus the targets have more than 1.2x down regulation in *mell* sample than WT sample but were not cleaved in *mell* sample were then filtered out. The rest of the predicted targets are considered as the targets of reproductive phasiRNAs. Indeed, the expression changes can only be used to help narrow the list of putative targets. We have added detailed description about how to filter targets using expression data and rephrased the description according to the targets expression pattern in WT samples in the revised manuscript (**page 6, line 126-131; page 7, line 134, marked in blue**), thank you for the indication. The figure legend of Fig 1d was also revised as suggested.

Comment 6. I am also concerned about potential bias in Figure 1e. It is not clear to me how cleavage position factors into sPARTA's prediction of target genes. Are PARE reads showing cleavage at a non-standard position weighed the same as PARE reads with expected cleavage? If not, then Figure 1e is biased in the same way as Figure 1d.

Reply: Thank you for the indication. We have consulted to the author of sPARTA

about the cleavage site bias of the arithmetic. Non-standard positions do not weigh the same with 9th, 10th and 11th of the binding site, thus we have deleted Fig 1e and the related sentences as suggested. Furthermore, we have performed additional RACE experiments to verify the cleavage sites, and the results supported that the cleavages preferred to occur between 9th, 10th and 11th of the phasiRNA binding sites (**Fig 3b**). Thank you again for your thoroughly review.

Comment 7. Lines 135-144: This section is confusing. Because the citations are numerical, I have to look at the reference list to even figure out which paper contains this “data from maize”. It would be much better to explain in the main text what this data is and how it was analyzed, particularly as the cited reference concludes that “clustering removed information about developmental dynamics” and they abandoned this method. Therefore when this paper attempts to overlap GO enrichment terms between “reported cluster 1” and their rice phasiRNA targets, I have no idea why that is valid. To be fair, I did not dig through the supplemental materials for the cited paper, but should I have to in order to understand this one?

Reply: Sorry for not explaining these data more clearly in original version of the manuscript. In the ref1⁶, the authors applied the method “clustering to group cells with similar gene expression patterns” which “removed information about developmental dynamics to identify intermediated stages from sequencing data” and then abandoned. Finally, they introduced a quantitative framework “pseudotime velocity” to identify cellular intermediates. After that, the authors clustered differentially expressed genes and grouped them to analyze the enriched GO term by AgriGO of each group⁶.

In this study, we analyzed the enriched GO term by AfriGO of the phasiRNA targets. Then the enriched F GO terms were compared with the F GO terms enriched in the reported gene groups. To make it more clearly, we have labeled the overlapped F GO terms in the revised **DataS4** and rephrased the manuscript as suggested (**page 7, line 143-145, marked in blue**), thank you.

Comment 8. Figure 2c-e: for the transgenic lines, wild-type is not a sufficient control. Over-expression of a non-phasiRNA targeted gene (but critically, a gene expressed during pollen development!!) is required. This experiment just shows that genes selected for pollen expression (and possibly selected for dynamic pollen expression) need to be correctly expressed for normal pollen development. (Or even worse, it shows that the transformation process impacts fertility.) An even better experiment would be to express these genes by their native promoter but with silent mutations that eliminate phasiRNA binding.

Reply: To address the comment, we have done following experiments in the revision: (1) we obtained the overexpression transgenic plants of non-phasiRNA targeted genes, including two laccase genes that express in anther (LAC10 and LAC17) and two other genes that do not express in anther (LAC20 and LAC23). We used these laccase genes for comparison because we previously found a laccase gene LAC13 involved in male fertility⁷. These overexpression transgenic plants were constructed using the same overexpression vector and transformation method together with phasiRNA target overexpressing plants. As shown in below Figure 4, these control plants have no obvious phenotype during pollen development, indicating that the pollen defects of phasiRNA target overexpression plants is not caused by the transformation of overexpressing vectors. We have provided two of them as controls in the revised supplementary **Fig S5b-e** as suggested (**page 9, line 177-179, marked in blue**), thank you.

(2) To further support the role of phasiRNA and MEL1 on target gene abundance, we have taken your suggestion by expressing target genes with silent mutations that can eliminate phasiRNA binding and performed additional experiments using rice protoplast system. Two target genes (LOC_Os08g35600 and LOC_Os03g43720) were used for validation. Rice protoplasts were transiently expressed the target genes or target gene with mutations that eliminate phasiRNA binding, together with or without

MEL1 and/or phasiRNA respectively. The result showed that the downregulation of target gene required the existing of MEL1 and phasiRNA, and mutation of the phasiRNA binding site suppressed the downregulation, which further supported the regulatory role of phasiRNA and MEL1 on target gene abundance. The result has been added to the revised **Fig 3a** and manuscript (page 8, line 174-177, marked in blue), thank you.

Figure 4. Transformation controls. **a**, Expression patterns of four control genes (LAC10, LAC17, LAC20 and LAC23) and two phasiRNA target genes during pollen development; **b**, Relative expression level of overexpression transgenic plants; **c**, Pollen grains of four control transgenic plants. Scale bars = 100 μ m; **d**, Seed setting rates of four control transgenic plants.

Comment 9. The conclusion of this section of results is that “MEL1 is indispensable for reprogramming of mRNA expression during early meiosis.” If I understand correctly, this conclusion is based on overlap between GO terms enriched among phasiRNA targets and genes upregulated in *mell1*, or genes shown to vary during maize development. I am again concerned that gene expression changes were used to *define* the phasiRNA targets, making this logic circular.

Reply: In this section, the conclusion was obtained from the observation that in melli plants, there were much more upregulated genes (including but not limited to phasiRNA targets) than downregulated genes (including but not limited to phasiRNA targets) when compared with WT plants (**Fig 2b**). In addition, phasiRNAs target to a number of transcription and RNA metabolism related genes. To make the statement more clearly, we have rephrased the related sentences as suggested (**page 8, line 167-168, marked in blue**), thank you.

Comment 10. I have a similar concern for experiments shown in Fig 5b-c and Fig 6. The appropriate control for these edited plants are heterozygous or homozygous WT siblings from a segregating population. Otherwise there is no control for the effects of transgenesis or other potential changes in the background.

Reply: We have used both transgenic plants that transferred with empty vector and wild type plants as the control for overexpression transgenic plants. For the CRISPR-Cas9 edited mutants, we have used both transgenic plants that transferred with CRISPR-Cas9 vectors but have no edition in the target sites and wild type plants as the control. We have provided more information and revised the methods section of the manuscript to make it more clear (**page 19, line 414-419, marked in blue**), thank you.

Comment 11. Figure 3C is an important panel to verify the PARE data, which otherwise has no replication. Please explain this panel better and make fonts larger to be more legible. Is “seq-40774” a phasiRNA? Does the arrow with “5/7” indicate 5 of 7 cleavage products were found at that site? Also how were these three targets selected? Was RLM-RACE attempted with additional targets, or were these the only successful examples?

Reply: The RACE experiments have three biological replicates. In response to your comment and the comment from reviewer 2, we have performed additional experiments to validate more target genes by RACE experiments (**Fig 3b**). Fig 3b and its figure legend were then revised to make it legible as suggested (“seq-40774” is a phasiRNA, and “5/7” indicate 5 of 7 cleavage products were found at that site). We have also rephrased the related sentences about the selection of targets (**page 9, line 196-199, marked in blue**). Briefly, both cleavage sites that have high number of PARE reads (15~180) and that have low PARE reads (< 10) were chosen randomly for RACE validation, and 10 of 15 chosen cleavage sites were successfully validated by RLM-RACE, thank you.

Comment 12. The description of a mi2118 mutant in rice is interesting, however as this line was provided by another laboratory, has it been published elsewhere? If not, please provide full details of how it was created (sequence of STTM, for example). Also, how was miR2118 expression quantified in Figure S5? The methods include qRT-PCR of phasiRNAs, but not miRNAs.

Reply: We agree. The mutant is not published elsewhere, and thus we had provided details of the mutant and the method of miR2118 detection in the revised methods section as suggested (**page 19, line 405-409, marked in blue**), thank you.

Comment 13. In Figure 4b, please report how many pollen grains were assessed at each anther length, how many anthers per size class, and how many plants those anthers were derived from. The methods state that at least 30 plants were analyzed, but it's not clear which analysis this refers to. As an aesthetic point, I find this type of 3D bar graph very difficult to interpret due to bars overlapping, perspective distorting sizes, etc.

Reply: More than 200 pollen mother cells were accessed from 30 ~ 40 anthers for each anther size class, and they were collected from five *dcl4* transgenic plants and

ten WT plants. The information was added to the revised figure legend of **Fig 4b**. **Fig 4b** was also adjusted to 2D graphics as suggested to make it more legible, thank you.

Comment 14. In Figure 4C it is not possible to see the red arrows that indicate chromosomal abnormalities, nor are these images of sufficient quality to identify those changes myself.

Reply: We have improved the quality of **Fig 4c** as suggested, thank you.

Comment 15. Seeing some inviable pollen (Fig 6a) is not sufficient to determine that “decreased fertility is caused by aborted pollen grains”. Plants make far more pollen than they need and it is common to have pollen defects with no impact on fertility. The correct experiment is reciprocal crosses between *msppl* mutants and WT to demonstrate that seed set is controlled by paternal genotype.

Reply: Thank you for the suggestion. To address this concern, we have crossed transgenic plants with WT plants to prove that the defects of pollen grains in these mutants impaired pollination and seed setting. For detail please see the reply of your comment 2. The result has been added to the revised **Fig S6a, b** and manuscript (**page 9, line 183-188; page 11, line 237-238, marked in blue**), thank you.

Comment 16. The conclusions in lines 226-229 are not supported by the data. I cannot conclude that “sequences downstream from the miR2118-binding site are more important than those upstream” when there are no direct comparisons between these - only comparisons with WT. Even if that comparison were made, it would only apply to the PHAS locus for which there is data. More examples are needed to make a generalization about all PHAS loci.

Reply: We have added the comparison between different lines as suggested (**Fig S7b**). We have also rephrased the sentences to “the sequences downstream from the

miR2118-binding sites in *MSPPL1* and *MSPPL2* are more important than those upstream and that the degree of sequence integrity at these two *PHAS* loci is essential for rice male fertility” to make it more precise (**page 12, line 244-246, marked in blue**), thank you.

Comment 17. It is difficult to interpret the data in Fig 6b-e without a schematic of the *PHAS* locus that shows where these phasiRNA sequences are generated in relation to the miR2118 site and the edits. Are the two “edited” phasiRNAs in Fig6b experimentally confirmed through sequencing? If not, how was their abundance assessed? The legend says that these are “edited” in the *mssp11* mutant, but at least three different alleles for *mssp11* are described - are they predicted to be created in all alleles?

Reply: We have added a schematic of the *PHAS* locus as suggested (**Fig S7b**). The two edited phasiRNAs were not existed in WT samples. We have examined the primers for detecting these two edited phasiRNAs, and performed more biological replicates, the results showed significant elimination of endogenous phasiRNAs and accumulation of edited phasiRNAs in the mutants. As their sequences are similar, the primers had a slightly nonspecific amplification. We have added the new data to the revised **Fig S8a**, thank you.

Comment 18. Figure 6c show *mssp11-1* and *mssp11-2*, which I gather are the two independent lines of *mssp11* described in lines 243-244. Both are “PAM2-edited”, but does that mean they are siblings with the same allele, or does it mean they are independent edits from the PAM2 transgenic line? Throughout this section, I have a lot of confusion about the different alleles created by these edits and would like to see the full genotypes of all plants.

Reply: *mssp11-1* and *mssp11-2* are two independent lines with the same edition at PAM2 (which affects the phasiRNA sequences), and different edition at PAM1 (which

do not affect the phasiRNA sequences). We have provided the genotypes of all the transgenic lines of both *msppl1* and *msppl2* which were used in this study as suggested (**Fig S7b**). Thank you.

Comment 19. Figure 6 needs statistical testing and confirmation that the replicates were independent biological replicates (ideally, different plants) and not technical replicates.

Reply: The qRT-PCR analysis all had three biological replicates. We have added the statistical testing to the revised **Fig 6** (Fig 6b and c were moved to the revised **Fig S8**) as suggested, thank you.

Comment 20. The data in Figure S5b don't make sense to me. Isn't deletion of the miR2118 expected to cause an increase in the PHAS transcript by eliminating its conversion to phasiRNAs?

Reply: We have used a primer pair which partially overlapped with the deletion site (primer 2 in the Figure 5 below), thus the abundance of PHAS transcript was underestimated in miR2118 deletion mutant. To analyze the accurate abundance of PHAS transcript, we further design three other primer pairs and examined the PHAS expression level (please see below the Figure 5). The result showed that the PHAS transcript was upregulated in the *msppl1*-miR2118 site deletion mutant (primer 3 and primer 4). We have used the primer 3 in the revised **Fig S7c**, thank you for the indication.

Figure 5. Relative expression level of *MSPPL1* in *msppl1*-miR2118 site deletion mutants. Four primer pairs were used to amplify *MSPPL1* in the *msppl1*-miR2118 site deletion mutant.

Comment 21. For dataset 1, please provide the number of mapped reads in addition to total reads.

Reply: The number of mapped reads has been provided in the revised data S1, thank you.

Comment 22. Figure S3a needs more description - what is the scale, how were samples normalized?

Reply: We have revised the figure legend of Fig S3a as suggested, thank you.

Comment 23. In Figure 1f I assume “reproductive” phasiRNAs were those identified in this study. What are the “other” phasiRNAs?

Reply: Reproductive phasiRNAs were those highly expressed in PFS and significantly decreased ($p < 0.05$) at EMS. The other phasiRNAs were referred as those upregulated in EMS or continuously expressed from PFS to EMS, which might play a

role during late sporogenesis.

Comment 24. Figure 3A is not a useful panel in the manuscript. And Figure 3b is a model that is more meaningful as a final figure.

Reply: We agree and have moved Fig 3a to **supplementary Fig 6c**, and moved Fig 3b to **Fig 6d** as suggested, thank you.

Comment 25. How was “seed setting rate” determined (Fig 5c)? Is this based on total seed number? Seed weight? Rate of plants that set any seed at all?

Reply: Seed setting rate determined was according to previously reported which indicates spikelet fertility and was estimated as the ratio of number of filled grains to total number of florets in a panicle^{8,9}. We have added the description to the revised methods section (**page 19, line 411-412, marked in blue**). Thank you again for all your comments.

1. Dai, P. et al. A Translation-Activating Function of MIWI/piRNA during Mouse Spermiogenesis. *Cell* **179**, 1566-1581 e16 (2019).
2. Das, S., Swetha, C., Pachamuthu, K., Nair, A. & Shivaprasad, P.V. Loss of function of *Oryza sativa* Argonaute 18 induces male sterility and reduction in phased small RNAs. *Plant Reprod* **33**, 59-73 (2020).
3. Araki, S. et al. miR2118-dependent U-rich phasiRNA production in rice anther wall development. *Nat Commun* **11**, 3115 (2020).
4. Brodersen, P. et al. Widespread translational inhibition by plant miRNAs and siRNAs. *Science* **320**, 1185-90 (2008).
5. Lanet, E. et al. Biochemical evidence for translational repression by Arabidopsis microRNAs. *Plant Cell* **21**, 1762-8 (2009).
6. Nelms, B. & Walbot, V. Defining the developmental program leading to meiosis in maize. *Science* **364**, 52-56 (2019).
7. Yu, Y. et al. Laccase-13 Regulates Seed Setting Rate by Affecting Hydrogen Peroxide Dynamics and Mitochondrial Integrity in Rice. *Front Plant Sci* **8**, 1324 (2017).
8. Li, S. et al. Natural variation in PTB1 regulates rice seed setting rate by controlling pollen tube growth. *Nat Commun* **4**, 2793 (2013).

9. Prasad, P.V.V., Boote, K.J., Allen, L.H., Sheehy, J.E. & Thomas, J.M.G. Species, ecotype and cultivar differences in spikelet fertility and harvest index of rice in response to high temperature stress. *Field Crops Research* **95**, 398-411 (2006).

REVIEWER COMMENTS

Reviewer #1 (Remarks to the Author):

Dear authors and editor,

To be honest, I was more impressed with the initial submission than with this revision. Although most reviewer comments were addressed, they were not always done so in a sufficient way which partly seems to be due to incomplete understanding of some raised points.

To be fair, this might be partly due to language difficulties which are also apparent in the spelling and grammar throughout the text and again (maybe even more so) in the added parts in the revision. I was disappointed that the language edits (partly also improving the statements) from my commented pdf were not included at all. Please do so, and know that I did not edit anything additionally although there are many instances where the word choice and sentences make understanding difficult (e.g. Fig 2 "Catalogs" -> "Categories").

Still, the revision content clearly improved the manuscript, for example by making figures clearer and explaining underlying methods and plant material.

Here my few specific comments:

- Please go through my previous grammar and spelling editing and add it in (see commented pdf document), also my other comments in the pdf, e.g. Fig5a: Were MSPPL1 targets supposed to be in the figure? If not, the MSPPL1 part is not very informative.
- Several reviewer comments touched on the data by Komiyama et al 2014 for MEL1-bound phasiRNAs. It would have been a neat additional analysis to use their data and compare it with this study. E.g. overlap of MEL1-bound phasiRNAs and MEL1-dependent phasiRNAs identified here by using mel1 transcriptome data in the definition of phasiRNAs of interest. Also, are the phasiRNAs from the phasi-RNA-target pairs MEL1-bound in Komiyama's data?
- Fig S8: Do not use asterisks to indicate mutated versions; asterisks should be used only for significance, especially if done so in another part of the figure
- Fig S6: Proper controls would be reciprocal crosses (also using WT as male, and mutants as female)
- Line 197ff: what about the 5 without shown cleavage sites? Were they of the <10 PARE reads category?
- Line 327-331: the previous statement was clearer, although the mentioning of gene family targeting adds value. The corresponding reviewer comment however was about something else: That randomly chosen sRNA will also often be predicted to target several genes – here however, you had degradome data to strengthen your finding.

Again, please revisit the notes from my (and maybe others?) first commented pdfs, and think about improving the language to make the topic and results which I deem highly valuable better to grasp for your future readers.

Reviewer #2 (Remarks to the Author):

The revised manuscript provided by Zhang et al is indeed a better version than the original one, with most of my concerns being properly addressed. I do have still some comments/suggestions that I believe need to be taken in consideration before publication of the manuscript.

- When selecting for putative phasiRNA targets (lines 126-131) the authors excluded all targets for which expression have more than 1.2x down regulation in mel1 sample. Why not to exclude all transcripts with downregulation? As the authors state earlier on (lines 126-127) "mRNA abundance in samples that the mRNAs were cleaved should not be higher than those have no cleavage".

- On the same line, if you believe that another AGO could be responsible for the target cleavage, wouldn't be better not to exclude any of the mel1 downregulated targets?

Or even better, have both information, transcripts that are Mel1 dependent and independently

regulated.

- Regarding the experiment where the phasiRNA precursors are mutated (lines 225-233). Although the authors have now included more information, it is still a bit hard to follow and to understand completely the reasons behind each mutants. I think the authors could spend a bit more time introducing and describing the mutants in the main text. For example, they mentioned they want to mimic natural variations that might occur elsewhere. Is there a specific one in mind (as indicated in the discussion section)? Why using specifically MSPPL1/2? Were the sites up- and downstream of miR2118 chosen for some reason? Which were the mutants generated, what are their characteristics. For example, does simultaneous PAM1+PAM2 editing cause deletion of the miR2118?

- In lines 258-259, the authors state that mutations downstream to the miR2118 can generate new phasiRNAs, which would have new targets. Although this is true and expected, it would be more interesting to do a similar approach to the one done for MSPPL2, and study how these mutations affect the expression of transcripts targeted by the original phasiRNAs. This analysis is much more meaningful for the scope of the paper, since it corroborates the role of these phasiRNA in regulating genes and in the plant development.

- In lines 268-271, the authors describe an experiment to study the putative targets of four phasiRNAs originating from MSPPL2, for which sequence has been affected by mutations. These targets are part of the 7 "randomly" selected putative targets tested in figure 2. Is this a coincidence, or, differently from what has been stated earlier on, the selection of these targets was not random (line 172)?

- In line 306, the authors claim that the targeting rules of these phasiRNAs are significantly different from other plant sRNAs. I believe this is an overstatement, and it would need further analysis from the authors. It is true that most miRNAs show a high degree of complementarity to their targets. This is most likely due to evolutionary forces and the key role of miRNAs in the development. However, there are cases where miRNAs can cause downregulation of transcripts even when several mismatches between the miRNA and the target exists (for ex, overexpression of miR319 can lead to downregulation of targets having even 5 mismatches, Palatnik et al 2003). Also, siRNAs can produce significant off-targeting due to less complementary targeting (Jackson et al 2003). It is highly possible that this tolerance to mismatches that phasiRNAs/targets pair seem to have when compared to miRNAs is not due to different targeting rules, but just less selective pressure. For instance, it is conceivable that sRNAs that have less target complementary are less efficient in downregulating transcripts. In the case of mRNAs targeted by just one sRNA, this scenario is not optimal, therefore, a selective pressure favoring a high amount of complementarity might exist. In the case of transcripts targeted by two phasiRNAs, this situation is not so important, since the double targeting could compensate for the decreased efficiency. Thus, it is likely that those phasiRNAs would not suffer the same evolutionary pressure than other sRNAs that are the sole targeting molecules. In conclusion, it is not the targeting rules that are different, but how these rules are enforced.

Minor comments:

- The blue color in the figures could be a brighter blue, to better differentiate from black.

- In line 255, the authors state that "Deletion of the miR2118-binding site downregulated MSPPL1 expression (Fig S8a)". However, in figure S8a only the MSPPL1-dependent phasiRNAs are shown, but not the precursor.

- I think it would be better to merge Figure 6a, b, c to Figure 5, since they are all analyzing the mutations in the MSPPL precursor. Also, the name of the 4 phasiRNAs analyzed in figure 6b and 6c could be added to the illustration in Fig 5a. The new figure 6 would be formed just by the scheme illustrated in Figure 6d.

- In Figure 6d, it would be better to place the legend indicating the phasiRNA and the mRNA expression on the right side of the graph, to make it clear it refers to that part of the illustration.

- There are a few grammatical/spelling mistakes that need to be corrected.

Reviewer #3 (Remarks to the Author):

The authors have greatly improved their manuscript and satisfied this reviewer sufficiently. My only remaining suggestions refer to figures.

Some of the text in figures is still very small (eg, Fig 2, Fig S4), although perhaps they will be reproduced larger in the final production. Since Nature Communications is not a print journal and therefore not limited by page number, the authors might consider increasing the size of some figures. Fig 4C and S4, especially.

Also, Figure 1E needs a label on the y-axis

REVIEWER COMMENTS

Reviewer #1 (Remarks to the Author):

Comment 1: To be honest, I was more impressed with the initial submission than with this revision. Although most reviewer comments were addressed, they were not always done so in a sufficient way which partly seems to be due to incomplete understanding of some raised points.

To be fair, this might be partly due to language difficulties which are also apparent in the spelling and grammar throughout the text and again (maybe even more so) in the added parts in the revision. I was disappointed that the language edits (partly also improving the statements) from my commented pdf were not included at all. Please do so, and know that I did not edit anything additionally although there are many instances where the word choice and sentences make understanding difficult (e.g. Fig 2 “Catalogs” -> “Categories”).

Still, the revision content clearly improved the manuscript, for example by making figures clearer and explaining underlying methods and plant material.

Reply: First of all, we would like to express our deeply apology for not realizing the editing and comments you marked in the attached PDF documents in previous version. We feel very sorry on this mistake. Thank you for your thoroughly review. In this version, we have carefully gone through the previous commented pdf document and revised the manuscript accordingly. According to your comment, we have also used a language service and sent our manuscript for language editing to improve the spelling and grammar, thank you.

Comment 2: Here my few specific comments:

- Please go through my previous grammar and spelling editing and add it in (see commented pdf document), also my other comments in the pdf, e.g. Fig5a: Were

MSPPL1 targets supposed to be in the figure? If not, the MSPPL1 part is not very informative.

Reply: We have carefully gone through the previous editing and comments in the commented pdf document to revise the manuscript, thank you very much. *MSPPL1* targets have been added to **Fig 5a** as suggested, and their expression patterns in *msppl1* plants were also added to the revised **Fig S8b**.

Comment 3: - Several reviewer comments touched on the data by Komiya et al 2014 for MEL1-bound phasiRNAs. It would have been a neat additional analysis to use their data and compare it with this study. E.g. overlap of MEL1-bound phasiRNAs and MEL1-dependent phasiRNAs identified here by using *mel1* transcriptome data in the definition of phasiRNAs of interest. Also, are the phasiRNAs from the phasi-RNA-target pairs MEL1-bound in Komiya's data?

Reply: Following your suggestion, we have performed the identification of MEL1-bound phasiRNAs using the the data by Komiya et al 2014 and overlapped the MEL1-bound phasiRNAs with the reproductive phasiRNAs that we identified in this study. The result showed that 3,516 reproductive phasiRNAs were MEL1-bound. We have added the information of whether a phasiRNA has been detected to be MEL1-bound in the revised **Data S2** (a list of the reproductive phasiRNAs) and **Data S3** (a list of the phasiRNA-target pairs) as suggested.

Comment 4: - Fig S8: Do not use asterisks to indicate mutated versions; asterisks should be used only for significance, especially if done so in another part of the figure

Reply: We agree and have replaced asterisks to "ed." as suggested, thank you.

Comment 5: - Fig S6: Proper controls would be reciprocal crosses (also using WT as male, and mutants as female)

Reply: As suggested, we have added the results of crosses which used WT as male and mutants as female in the revised Fig S6a and manuscript (**page 9, line 195-196**, marked in blue). Together with the original results, when using these mutants as male parent, the seed setting rates decreased compared with that using WT as male parent or mutants as female parent (**Fig. S6a**), indicating that the defects of pollen grains in these mutants impaired pollination and seed setting. Thank you for the suggestion.

Comment 6: - Line 197ff: what about the 5 without shown cleavage sites? Were they of the <10 PARE reads category?

Reply: Yes, 3 of the 5 cleavage sites have PARE reads less than 10, and all the 5 genes have lower expression levels than most of the other 10 genes.

Comment 7: - Line 327-331: the previous statement was clearer, although the mentioning of gene family targeting adds value. The corresponding reviewer comment however was about something else: That randomly chosen sRNA will also often be predicted to target several genes – here however, you had degradome data to strengthen your finding.

Again, please revisit the notes from my (and maybe others?) first commented pdfs, and think about improving the language to make the topic and results which I deem highly valuable better to grasp for your future readers.

Reply: We agree with your comment and have rephrased the sentence accordingly (**page 17, line 356-361**, marked in blue). We have carefully read through the comments in the commented PDF documents and revised the manuscript accordingly. According to your comment, we have also used a language service to edit the manuscript to improve the spelling and grammar throughout the text. Thank you again for all your effort for improving the manuscript.

Reviewer #2 (Remarks to the Author):

Comment 1: The revised manuscript provided by Zhang et al is indeed a better version than the original one, with most of my concerns being properly addressed. I do have still some comments/suggestions that I believe need to be taken in consideration before publication of the manuscript.

- When selecting for putative phasiRNA targets (lines 126-131) the authors excluded all targets for which expression have more than 1.2x down regulation in mel1 sample. Why not to exclude all transcripts with downregulation? As the authors state earlier on (lines 126-127) “mRNA abundance in samples that the mRNAs were cleaved should not be higher than those have no cleavage”.

- On the same line, if you believe that another AGO could be responsible for the target cleavage, wouldn't be better not to exclude any of the mel1 downregulated targets? Or even better, have both information, transcripts that are Mel1 dependent and independently regulated.

Reply: Thank you for your further suggestion. For MEL1 dependent targets, we have excluded all transcripts downregulated as suggested (**Data S3**). The other cleaved targets which were filtered out and might be MEL1 independent targets were listed in the revised **Data S4** as suggested.

Comment 2: - Regarding the experiment where the phasiRNA precursors are mutated (lines 225-233). Although the authors have now included more information, it is still a bit hard to follow and to understand completely the reasons behind each mutants. I think the authors could spend a bit more time introducing and describing the mutants in the main text. For example, they mentioned they want to mimic natural variations that might occur elsewhere. Is there a specific one in mind (as indicated in the

discussion section)? Why using specifically MSPPL1/2? Were the sites up- and downstream of miR2118 chosen for some reason? Which were the mutants generated, what are their characteristics. For example, does simultaneous PAM1+PAM2 editing cause deletion of the miR2118?

Reply: We have added the description of the mutants in the main text as suggested, including the reported natural variations in *PHAS* loci (page 11, line 233-234, marked in blue), the reason for choosing *MSPPL1/2* (page 11, line 239-243, marked in blue) and the characteristics of different mutants (page 12, line 251-260, marked in blue). Thank you for the advice.

Comment 3: - In lines 258-259, the authors state that mutations downstream to the miR2118 can generate new phasiRNAs, which would have new targets. Although this is true and expected, it would be more interesting to do a similar approach to the one done for MSPPL2, and study how these mutations affect the expression of transcripts targeted by the original phasiRNAs. This analysis is much more meaningful for the scope of the paper, since it corroborates the role of these phasiRNA in regulating genes and in the plant development.

Reply: Following your suggestion, we have analyzed the expression level of target genes of phasiRNAs generated from *MSPPL1* in different *mस्पpl1* plants as suggested. The two targets were up-regulated in *mस्पpl1*. The result was added to **Fig. S8b** and the revised manuscript (page 13, line 283-284, marked in blue). Thank you.

Comment 4: - In lines 268-271, the authors describe an experiment to study the putative targets of four phasiRNAs originating from MSPPL2, for which sequence has been affected by mutations. These targets are part of the 7 “randomly” selected putative targets tested in figure 2. Is this a coincidence, or, differently from what has been stated earlier on, the selection of these targets was not random (line 172)?

Reply: Sorry for the unclear description. Fig 2c-e showed the phenotypes of both the seven randomly chosen target genes and the target genes of phasiRNAs originating from *MSPPL2*. To make it more clear, we have moved the results of target genes of phasiRNAs originating from *MSPPL2* to the revised **Fig 5g-i**. Thank you for the indication.

Comment 5: - In line 306, the authors claim that the targeting rules of these phasiRNAs are significantly different from other plant sRNAs. I believe this is an overstatement, and it would need further analysis from the authors. It is true that most miRNAs show a high degree of complementarity to their targets. This is most likely due to evolutionary forces and the key role of miRNAs in the development. However, there are cases where miRNAs can cause downregulation of transcripts even when several mismatches between the miRNA and the target exists (for ex, overexpression of miR319 can lead to downregulation of targets having even 5 mismatches, Palatnik et al 2003). Also, siRNAs can produce significant off-targeting due to less complementary targeting (Jackson et al 2003). It is highly possible that this tolerance to mismatches that phasiRNAs/targets pair seem to have when compared to miRNAs is not due to different targeting rules, but just less selective pressure. For instance, it is conceivable that sRNAs that have less target complementary are less efficient in downregulating transcripts. In the case of mRNAs targeted by just one sRNA, this scenario is not optimal, therefore, a selective pressure favoring a high amount of complementarity might exist. In the case of transcripts targeted by two phasiRNAs, this situation is not so important, since the double targeting could compensate for the decreased efficiency. Thus, it is likely that those phasiRNAs would not suffer the same evolutionary pressure than other sRNAs that are the sole targeting molecules. In conclusion, it is not the targeting rules that are different, but how these rules are enforced.

Reply: Thank you for your comments. We agree with your comment. We have deleted “targeting rules” and rephrase the sentence to “This effect might be due to different

evolutionary pressures on reproductive phasiRNAs and miRNAs during development”
(page 16, line 345-346, marked in blue).

Comment 6: Minor comments:

- The blue color in the figures could be a brighter blue, to better differentiate from black.

Reply: We have changed the blue color to a brighter blue in the revised **Fig. 3** and **Fig. S7** as suggested.

Comment 7: - In line 255, the authors state that “Deletion of the miR2118-binding site downregulated MSPPL1 expression (Fig S8a)”. However, in figure S8a only the MSPPL1-dependent phasiRNAs are shown, but not the precursor.

Reply: Sorry for the mistake. We have revised the sentence and cited the right figures (**Fig. S7c** and **S8a**). Thank you.

Comment 8: - I think it would be better to merge Figure 6a, b, c to Figure 5, since they are all analyzing the mutations in the MSPPL precursor. Also, the name of the 4 phasiRNAs analyzed in figure 6b and 6c could be added to the illustration in Fig 5a. The new figure 6 would be formed just by the scheme illustrated in Figure 6d.

Reply: Thank you for your suggestion. We have merged Fig. 6a, b, c to Fig. 5, and added the name of the phasiRNAs in **Fig. 5a** as suggested.

Comment 9: - In Figure 6d, it would be better to place the legend indicating the phasiRNA and the mRNA expression on the right side of the graph, to make it clear it refers to that part of the illustration.

Reply: We have moved the legend indicating phasiRNAs and mRNAs to the right side of the graph as suggested, thank you.

Comment 10: - There are a few grammatical/spelling mistakes that need to be corrected.

Reply: We have used a language service and sent our manuscript for editing to improve the spelling and grammar throughout the text. Thank you again for all your suggestions.

Reviewer #3 (Remarks to the Author):

Comment 1: The authors have greatly improved their manuscript and satisfied this reviewer sufficiently. My only remaining suggestions refer to figures.

Some of the text in figures is still very small (eg, Fig 2, Fig S4), although perhaps they will be reproduced larger in the final production. Since Nature Communications is not a print journal and therefore not limited by page number, the authors might consider increasing the size of some figures. Fig 4C and S4, especially.

Reply: Thank you for your comments. We have enlarged the text in **Fig. 2** and **Fig. S4** as suggested. We have also increased the size of all the figures.

Comment 2: Also, Figure 1E needs a label on the y-axis

Reply: We have added the label of y-axis in **Fig. 1e** as suggested. Thank you again for all your suggestions.

REVIEWERS' COMMENTS

Reviewer #1 (Remarks to the Author):

The authors now did a truly laudable effort to polish the manuscript by adding all provided edits, using a language service and reacting and resolving all reviewer issues. I thus completely endorse the manuscript as it is now.

Very small comments for the final touch are:

- Fig S6: The usual convention is to write the female first and then the male parent.
- P17: Thanks for taking my input to heart; to make the statement again simpler and clearer, you could replace "even though phasiRNAs or randomly chosen sRNAs in plants are predicted to target several genes when a greater degree of mispairing is tolerated" by: "since other plant sRNAs usually target only one gene or several genes of the same gene family". Also, you could then remove the added "based on relatively low sequence complementary". For me, my main point was for you to understand that multiple targeting is easily predicted, but in your case also verified.
- P9 line 195: "or the mutants as female parents" \diamond "and the mutants as female parents"
- In Fig 6, the phasiRNA + mRNA amount picture in prophase is a bit confusing due to the (Western) reading direction from left to right. I would suggest to flip it vertically, and putting the label "Prophase I \diamond " on the left side of it, writing from top to bottom.

Reviewer #2 (Remarks to the Author):

The modifications made by the authors have significantly improved the manuscript with most of my concerns being properly addressed. However, I do think the points below need to be addressed before publication:

Line 251-260: With the newly added description of the lines it is much clearer now to understand the respective experiment, but it would be a good idea to specify the name of each line in accordance to the figures. For ex: 1) Deletion of the miR2118 binding site (miR2118 del); 2) edited sequences upstream from the potential miR2118 binding site (but that did not affect the phasiRNA sequence or miR2118-binding site) (PAM1&PAM2 edit.1 and PAM1&PAM2 edit.2), etc... Also, it would be interesting to indicate which line was used for the experiments depicted in Fig5. For instance, in Fig S7, two PAM1 edit lines exist for MSPPL1. Which one of them were used for the experiments shown in Fig 5c? The same for MSPLL2, which PAM2 4bp was used? (According to Figure S7, many of the different mutations have more than one line, with slightly differences).

Line 266-273 (and also discussion – lines 408-414). The authors concluded based on the seed setting rates that "sequences downstream from the miR2118-binding site in MSPPL1 and MSPLL2 are more important than those upstream". However, I don't agree that the authors can conclude this based on the data shown. Seed setting rates for msppl1 are pretty much the same. PAM1&PAM2 edit might indicate a stronger reduction, but also shows are much higher variation of phenotype, which I would be surprised if it will turn to be statistically (or biologically) significant. For msppl2 the situation is similar. The average rate does seem to be related to the size of deletion, but the variation in the phenotype (SD) is huge. In addition, all these deletions are in the PAM2 area, so there is no direct comparison with mutations occurring only in the PAM1 area. The same apply for msppl1, where only PAM1 or PAM1&PAM2 edit are compared, with no PAM2-only available for direct comparison. Therefore, the conclusion with that experiment is that mutations around the target site can affect the PHAS locus function.

In my opinion, the authors should limit their conclusion to their final sentence: "the degree of sequence integrity at these two PHAS loci is essential for rice male fertility". Maybe they could include a small discussion on why mutations around the target site have such an effect on the phasiRNAs, making parallels to what has been described by Guan et al DOI: 10.1038/ncomms4050

Minor comment:

- Line 123: I believe after <0.05 it should be a ";" and not ":".

Reviewer #1 (Remarks to the Author):

Comment 1. The authors now did a truly laudable effort to polish the manuscript by adding all provided edits, using a language service and reacting and resolving all reviewer issues. I thus completely endorse the manuscript as it is now.

Very small comments for the final touch are:

- Fig S6: The usual convention is to write the female first and then the male parent.

Reply: We appreciate the comment. We have revised Fig. S6 as suggested, thank you.

Comment 2. P17: Thanks for taking my input to heart; to make the statement again simpler and clearer, you could replace “even though phasiRNAs or randomly chosen sRNAs in plants are predicted to target several genes when a greater degree of mispairing is tolerated” by: “since other plant sRNAs usually target only one gene or several genes of the same gene family”. Also, you could then remove the added “based on relatively low sequence complementary”. For me, my main point was for you to understand that multiple targeting is easily predicted, but in your case also verified.

Reply: We have revised the sentences as suggested (page 17, line 357-358, marked in blue), thank you.

Comment 3. P9 line 195: “or the mutants as female parents” □ “and the mutants as female parents”

Reply: We have replaced the “or” as “and” as suggested (page 9, line 196, marked in blue).

Comment 4. In Fig 6, the phasiRNA + mRNA amount picture in prophase is a bit

confusing due to the (Western) reading direction from left to right. I would suggest to flip it vertically, and putting the label “Prophase I □” on the left side of it, writing from top to bottom.

Reply: We have revised Fig. 6 as suggested, thank you again for all your suggestions.

Reviewer #2 (Remarks to the Author):

Comment 1. The modifications made by the authors have significantly improved the manuscript with most of my concerns being properly addressed. However, I do think the points below need to be addressed before publication:

Line 251-260: With the newly added description of the lines it is much clearer now to understand the respective experiment, but it would be a good idea to specify the name of each line in accordance to the figures . For ex: 1) Deletion of the miR2118 binding site (miR2118 del); 2) edited sequences upstream from the potential miR2118 binding site (but that did not affect the phasiRNA sequence or miR2118-binding site) (PAM1&PAM2 edit.1 and PAM1&PAM2 edit.2), etc...

Reply: Thank you for your further suggestion. We have specified the name of each line as suggested (**page 12, line 253, 255, 257 and 262**, marked in blue; also in **Fig. S7**), thank you.

Comment 2. Also, it would be interesting to indicate which line was used for the experiments depicted in Fig5. For instance, in Fig S7, two PAM1 edit lines exist for MSPPL1. Which one of them were used for the experiments shown in Fig 5c? The same for MSPLL2, which PAM2 4bp was used? (According to Figure S7, many of the different mutations have more than one line, with slightly differences).

Reply: We have added the line information to the Fig. 5(b-f) and its legend as suggested, thank you.

Comment 3. Line 266-273 (and also discussion – lines 408-414). The authors concluded based on the seed setting rates that “sequences downstream from the miR2118-binding site in MSPPL1 and MSPPL2 are more important than those upstream”. However, I don’t agree that the authors can conclude this based on the data shown. Seed setting rates for msppl1 are pretty much the same. PAM1&PAM2 edit might indicate a stronger reduction, but also shows are much higher variation of phenotype, which I would be surprised if it will turn to be statistically (or biologically) significant. For msppl2 the situation is similar. The average rate does seem to be related to the size of deletion, but the variation in the phenotype (SD) is huge. In addition, all these deletions are in the PAM2 area, so there is no direct comparison with mutations occurring only in the PAM1 area. The same apply for msppl1, where only PAM1 or PAM1&PAM2 edit are compared, with no PAM2-only available for direct comparison.

Therefore, the conclusion with that experiment is that mutations around the target site can affect the PHAS locus function.

In my opinion, the authors should limit their conclusion to their final sentence: “the degree of sequence integrity at these two PHAS loci is essential for rice male fertility”. Maybe they could include a small discussion on why mutations around the target site have such an effect on the phasiRNAs, making parallels to what has been described by Guan et al DOI: 10.1038/ncomms4050

Reply: We have revised the conclusion and added a brief discussion in the discussion section as suggested (page 13, lines 273-274, page 19, lines 409 and 411-413, marked in blue), thank you.

Comment 4. Minor comment:

- Line 123: I believe after <0.05 it should be a “;” and not “:”.

Reply: We have corrected the punctuation (**page 6, line 124**, marked in blue). Thank you again for all your suggestions.